# Hyperbolic U-Net for Robust Medical Image Segmentation

**Swasti S. Mishra** [1]                                    S.S.MISHRA@UVA.NL
**Max van Spengler** [2]                          M.W.F.VANSPENGLER@UVA.NL
**Erwin Berkhout** [3]                                  E.BERKHOUT@ACTA.NL
**Pascal Mettes** [2]                                  P.S.M.METTES@UVA.NL

[1] *HAVA Lab, University of Amsterdam;* [2] *VIS Lab, University of Amsterdam;*

[3] *Academic Centre for Dentistry Amsterdam (ACTA), University of Amsterdam & Vrije Universiteit Amsterdam*

**Editors:** Accepted for publication at MIDL 2026

## Abstract

The U-Net architecture is a leading network in medical image segmentation. Despite its strong segmentation performance, U-Net struggles when dealing with noise in image data, such as random interference and brightness variations. While a common occurrence, the presence of random noise leads to strong performance degradation in U-Net, hampering its clinical integration and robustness. In this work, we investigate the role of geometry in U-Net. All U-Net variations share the same geometric foundations, namely Euclidean geometry. Here, we propose Hyperbolic U-Net, which maintains U-Net's proven encoder-decoder structure while operating entirely in the Poincaré ball of hyperbolic space. We identify two main roadblocks for training a fully Hyperbolic U-Net and propose a solution for each: (i) fully hyperbolic literature has so far focused on encoders, limiting their applicability to segmentation. We introduce hyperbolic 2D transpose convolution and hyperbolic bilinear upsampling layers that make it possible to create decoders, and (ii) existing hyperbolic parameter initializations are not suitable for hyperbolic decoder blocks. We introduce a Newton's approximation-scaled weight initialization, which ensures norm preservation for all layers at the start of training. Empirically, we show that our Hyperbolic U-Nets strongly outperform standard Euclidean U-Nets across multiple medical image datasets for Gaussian, Speckle, Poisson, and Rician noise, as well as to brightness and contrast shift. We conclude that a fully Hyperbolic U-Net is highly robust to out-of-the-box noise, without the need for denoising or additional objectives, highlighting the potential of hyperbolic geometry for medical imaging. Our code is available at https://github.com/swastishreya/Hyperbolic-U-Net.

**Keywords:** Medical Image Segmentation, Hyperbolic Neural Networks, Robustness

## 1. Introduction

Segmentation of images, typically defined as a per-pixel or per-region classification task, plays a crucial role in the medical domain, with the U-Net architecture as a ubiquitous neural network (Ronneberger et al., 2015). Since its inception, U-Net has become a staple in both academia and industry, due to its empirical effectiveness, modular design, and broad applicability to many data structures and tasks (Azad et al., 2024). Powered by continued advances in the design of network layers (Alom et al., 2019; Oktay et al., 2018; Zhou et al., 2019a) and optimization (Sudre et al., 2017; Isensee et al., 2021), U-Net has furthermore been extended to tasks such as image synthesis and reconstruction (Laxman et al., 2021; Wang et al., 2022; Sun et al., 2023).

A core remaining issue for image segmentation problems in the medical domain is dealing with noise. The data acquisition process of medical images is likely to induce artifacts and random noise (Nazir et al., 2024). Consider, for example, speckle noise, electronic noise, backscatter noise, radiation interference, scanner-specific artifacts, or simple variations between patients being captured (Kumar and Priyadarshi, 2025). The presence of such noise has a strong impact on the performance of U-Net (Cheng et al., 2024). As a consequence, constant re-training for every new scanner (Guo et al., 2024), prior denoising steps (Manakov et al., 2019), or additional algorithmic modules (Sheikh and Schultz, 2022) are needed to deal with noise. This limits the real-world integration of segmentation models and hampers scaling to larger data collections, which will invariably have multiple different noise properties.

In this work, we investigate the role of geometry for robustness in U-Net. Although many extensions and improvements have been introduced to U-Net, all share the same geometric foundations, namely Euclidean geometry. Recently, a wide range of works have shown that building deep learning in non-Euclidean geometries opens new doors for the field (Dhingra et al., 2018; Liu et al., 2019; Khrulkov et al., 2020; Ayubcha et al., 2024). Specifically, deep learning in hyperbolic space has shown strong potential for robustness, with hyperbolic neural networks being more resilient to adversarial attacks (van Spengler et al., 2024), improving out-of-distribution detection (Gonzalez-Jimenez et al., 2024), and naturally capturing uncertainty (Atigh et al., 2022), without the need for additional data or optimization. Based on these success cases, this work studies the potential of hyperbolic deep learning for noise robustness in medical image segmentation.

This work introduces Hyperbolic U-Net, the first U-Net fully developed in the Poincaré ball of hyperbolic space. In our approach, input images are first mapped to hyperbolic space, after which all layers of the encoder-decoder model are performed on a hyperbolic manifold. We propose hyperbolic 2D transposed convolution and hyperbolic bilinear upsampling layers, since fully hyperbolic literature has so far focused mainly on encoders only (Bdeir et al., 2024; Van Spengler et al., 2023). We furthermore introduce a Newton approximation-scaled weight initialization, which ensures norm-preservation of all layers at the start of training. We hypothesize that hyperbolic geometry can improve robustness to noise in medical image segmentation. Specifically, the exponential volume growth of hyperbolic space may increase relative separation between features corresponding to different semantic classes, which could reduce the likelihood that noise-induced perturbations cross class boundaries. By embedding features in this space, Hyperbolic U-Net is expected to maintain segmentation accuracy under a variety of noise conditions. To test this hypothesis, we perform extensive experiments across multiple datasets and noise types, comparing Hyperbolic U-Net to standard Euclidean U-Nets and other competitive baselines. Results show that the Hyperbolic U-Net is highly robust to many variants of random noise, with minimal impact in performance even at high noise ratios, whereas a standard U-Net and nnU-Net setups consistently struggle with noise. The results highlight the power of hyperbolic geometry for medical image segmentation.

## 2. Related Works

**Medical Image Segmentation.** U-Net (Ronneberger et al., 2015) is the foundation of modern medical image segmentation. Numerous variations have refined this backbone

through skip connections (Zhou et al., 2019a), attention mechanisms (Oktay et al., 2018), multi-scale feature extraction (Ibtehaz and Rahman, 2020), and residual (Milletari et al., 2016) or densely connected designs (Jégou et al., 2017). These advances consistently enhance feature representation and spatial detail. A persistent challenge in medical image segmentation is robustness to noise, artifacts, and acquisition variability (Nazir et al., 2024; Kumar and Priyadarshi, 2025). Several studies highlight that U-Net like networks experience substantial performance drops under noise perturbations or distribution shift (Kervadec et al., 2019; Alom et al., 2019; Valanarasu et al., 2021). Other studies expose the vulnerability of these networks to adversarial noise and provide robustness assessment frameworks (Asgari Taghanaki et al., 2018; Wang and Xu, 2024; Daza et al., 2021). To mitigate these issues, the community has explored stronger data augmentation (Zhou et al., 2019b), adversarial robustness training (Asgari Taghanaki et al., 2018; Wang and Xu, 2024), model architecture enhancement (Daza et al., 2021), defining new loss functions (Gonzalez-Jimenez et al., 2025b) and domain adaptation mechanisms such as unsupervised feature learning (Kamnitsas et al., 2017; Chen et al., 2020). Robustness can also be addressed through classical denoising objectives (Kaur et al., 2018; Wang et al., 2021), which requires separate training stages. While these methods improve stability, they typically require additional objectives, adversarial modules, or task-specific tuning. Holistic end-to-end robustness strategies such as nnU-Net (Isensee et al., 2018, 2021) attempt to reduce sensitivity to acquisition variability and noise through extensive normalization and augmentation policies. Despite such progress, U-Net remains sensitive to noise. This motivates us to explore an alternative geometry for U-Net that offers an inherent robustness to noise.

**Hyperbolic Neural Networks.** Owing to its strong ability to embed hierarchical and tree-like structures (Sarkar, 2011; Nickel and Kiela, 2017; Ganea et al., 2018a; van Spengler and Mettes, 2025), hyperbolic geometry has seen a surge in popularity within deep learning literature (Peng et al., 2021; Mettes et al., 2024), leading to numerous applications in NLP (Dhingra et al., 2018; He et al., 2024; Mandica et al., 2024), graph learning (Liu et al., 2019; Chami et al., 2019; Zhang et al., 2021) and, particularly, computer vision (Khrulkov et al., 2020; Guo et al., 2022; Ermolov et al., 2022; Desai et al., 2023; Poppi et al., 2025; Pal et al., 2025). Commonly, these approaches restrict the use of hyperbolic geometry to the final layers of the model. For example, Atigh et al. use a Euclidean segmentation model to obtain pixel-level features, which are projected to hyperbolic space and classified using a hyperbolic classifier. Hyperbolic learning has also shown great early potential in the medical domain. Ayubcha et al.; Gonzalez-Jimenez et al. map image features from Euclidean ResNets to the hyperboloid and perform medical classification. Similarly, Wang et al. map pixel features from a Euclidean vision transformer to the Poincaré ball and perform hyperbolic multi-lesion segmentation in diabetic retinopathy. Several approaches use the recent advances in hyperbolic multimodal learning to align the representations of medical data, such as images, brain activations or WSI hierarchies, with the representations from textual descriptions (Qiao et al., 2025; Pate et al., 2025; Huang et al., 2025; Baek et al., 2025). Current literature however focuses on hyperbolic geometry for the classification or semantic space only. This work strives to make the full architecture of U-Net hyperbolic. We note that the hyperbolic networks of Lensink et al. are based on the hyperbolic telegraph equation, which is unrelated to the hyperbolic geometry that is used in our work.

Ganea et al. proposed the first hyperbolic version of hyperbolic multinomial logistic regression on the Poincaré ball model, which was extended to a hyperbolic linear layer by Shimizu et al.. Chen et al. proposed an alternative architecture in the hyperboloid model of hyperbolic space. Early results with fully hyperbolic networks in image classification have shown improved robustness compared to their Euclidean and partially hyperbolic counterparts (Van Spengler et al., 2023; Bdeir et al., 2024). However, despite the importance of robustness in segmentation, no prior work has explored the potential of such architectures for this domain. We propose the first fully hyperbolic version of a U-Net for image segmentation and demonstrate that it achieves significantly increased robustness.

## 3. Methods

**Riemannian geometry.** Riemannian geometry is the study of Riemannian manifolds, which are pairs $(\mathcal{M}, \mathfrak{g})$ consisting of a smooth manifold $\mathcal{M}$ and a Riemannian metric $\mathfrak{g}$. Smooth manifolds are topological spaces that are locally similar enough to Euclidean space to allow the use of calculus on small neighborhoods, but which can differ tremendously from Euclidean space at larger scales. At each point $\boldsymbol{x}$ of $\mathcal{M}$, we can define the space of possible directions with velocities with which we can travel from $\boldsymbol{x}$ as $\mathcal{T}_{\boldsymbol{x}}\mathcal{M}$, which is often named the tangent space at $\boldsymbol{x}$. Locally, such a tangent space can be seen as the Euclidean approximation to the manifold at $\boldsymbol{x}$. The Riemannian metric $\mathfrak{g}$ is a function that assigns an inner product to each tangent space, which allows the computation of the lengths of curves defined on $\mathcal{M}$. Geodesics are special cases of such curves on manifolds, which generalize the concept of straight lines. When it exists, a geodesic between any two points $\boldsymbol{x}$ and $\boldsymbol{y}$ on the manifold is the curve that forms the shortest path between them.

Two important mappings on Riemannian manifolds are the exponential and logarithmic maps. The exponential map at some point $\boldsymbol{x}$ takes a tangent vector $\boldsymbol{v}$ from $\mathcal{T}_{\boldsymbol{x}}\mathcal{M}$ and outputs the point on the manifold that we would end up in if we were to travel from $\boldsymbol{x}$ in the direction of $\boldsymbol{v}$ with velocity $\sqrt{\mathfrak{g}(\boldsymbol{v}, \boldsymbol{v})}$ for 1 unit of time. The logarithmic map at $\boldsymbol{x}$ is the inverse of the exponential map, so it takes some point $\boldsymbol{y}$ and returns the tangent vector in $\mathcal{T}_{\boldsymbol{x}}\mathcal{M}$ that represents the direction and velocity needed to arrive at $\boldsymbol{y}$ in a single unit of time. In what follows, we will treat hyperbolic space as a Riemannian manifold and use the tools from Riemannian geometry to enable computation within this space. For a comprehensive overview of Riemannian geometry, we refer the reader to (Lee, 2018).

**Poincaré ball model.** This paper operates on the commonly used Poincaré ball model of hyperbolic space. For $n$-dimensional hyperbolic space with constant negative curvature $-c$, this is defined as the Riemannian manifold $(\mathbb{B}_c^n, \mathfrak{g}_c)$, where

$$\mathbb{B}_c^n = \left\{ \boldsymbol{x} \in \mathbb{R}^n : \|\boldsymbol{x}\|^2 < \frac{1}{c} \right\}, \quad \mathfrak{g}_c = (\lambda_{\boldsymbol{x}}^c)^2 I_n, \quad \lambda_{\boldsymbol{x}}^c = \frac{2}{1 - c\|\boldsymbol{x}\|^2}, \tag{1}$$

with $I_n$ being the $n$-dimensional identity matrix. The Poincaré ball model can be turned into a gyrogroup (Ungar, 2022) by endowing it with Möbius addition, defined as

$$\boldsymbol{x} \oplus_c \boldsymbol{y} = \frac{\left(1 + 2c\langle \boldsymbol{x}, \boldsymbol{y} \rangle + c\|\boldsymbol{y}\|^2\right) \boldsymbol{x} + \left(1 - c\|\boldsymbol{x}\|^2\right) \boldsymbol{y}}{1 + 2c\langle \boldsymbol{x}, \boldsymbol{y} \rangle + c^2\|\boldsymbol{x}\|^2\|\boldsymbol{y}\|^2}, \tag{2}$$

where $\boldsymbol{x}, \boldsymbol{y} \in \mathbb{B}_c^n$, $r \in \mathbb{R}$ and where $\|\cdot\|$ and $\langle\cdot, \cdot\rangle$ denote the Euclidean norm and the inner product, respectively. The exponential map projects a tangent vector back onto

the manifold along a geodesic, while the logarithmic map performs the inverse operation, mapping a point on the manifold to a tangent vector at a reference location. Using the definition of Möbius addition, the exponential and logarithmic maps can be written as

$$
\begin{aligned}
\exp_{\boldsymbol{x}}^c(\boldsymbol{v}) &= \boldsymbol{x} \oplus_c \left( \tanh \left( \frac{\sqrt{c} \lambda_{\boldsymbol{x}}^c \|\boldsymbol{v}\|}{2} \right) \frac{\boldsymbol{v}}{\sqrt{c} \|\boldsymbol{v}\|} \right), \\
\log_{\boldsymbol{x}}^c(\boldsymbol{y}) &= \frac{2}{\sqrt{c} \lambda_{\boldsymbol{x}}^c} \tanh^{-1} \left( \sqrt{c} \, \|-\boldsymbol{x} \oplus_c \boldsymbol{y}\| \right) \frac{-\boldsymbol{x} \oplus_c \boldsymbol{y}}{\|-\boldsymbol{x} \oplus_c \boldsymbol{y}\|},
\end{aligned}
\tag{3}
$$

where $\boldsymbol{x}, \boldsymbol{y} \in \mathbb{B}_c^n$ and $\boldsymbol{v} \in \mathcal{T}_{\boldsymbol{x}} \mathbb{B}_c^n$ (Ganea et al., 2018b). Furthermore, we can compute the distance between any two points $\boldsymbol{x}, \boldsymbol{y} \in \mathbb{B}_c^n$ as

$$
d_c(\boldsymbol{x}, \boldsymbol{y}) = \frac{2}{\sqrt{c}} \tanh^{-1} \left( \sqrt{c} \, \|-\boldsymbol{x} \oplus_c \boldsymbol{y}\| \right).
\tag{4}
$$

We follow Shimizu et al.; Van Spengler et al. to build a fully hyperbolic convolutional neural network. As a foundation, Poincaré multinomial logistic regression is defined by computing the score for each of $n$ classes for some $m$-dimensional input $\boldsymbol{x} \in \mathbb{B}_c^m$ as

$$
v_k(\boldsymbol{x}) = \frac{2}{\sqrt{c}} \|\boldsymbol{z}_k\| \sinh^{-1} \left( \lambda_{\boldsymbol{x}}^c \left\langle \sqrt{c} \boldsymbol{x}, \frac{\boldsymbol{z}_k}{\|\boldsymbol{z}_k\|} \right\rangle \cosh \left( 2\sqrt{c} r_k \right) - (\lambda_{\boldsymbol{x}}^c - 1) \sinh \left( 2\sqrt{c} r_k \right) \right),
\tag{5}
$$

where $\boldsymbol{z}_k \in \mathcal{T}_0 \mathbb{B}_c^m$ and $r_k \in \mathbb{R}$ are the parameters for the $k$-th class. These scores are equivalent to the distances between the input $\boldsymbol{x}$ and the $n$ different Poincaré hyperplanes determined by the parameters $\{(\boldsymbol{z}_k, r_k)\}_{i=1}^n$. Here, $\boldsymbol{z}_k$ determines the orientation of the hyperplane while $r_k$ determines its offset with respect to the origin. A Poincaré fully connected layer mapping input $\boldsymbol{x} \in \mathbb{B}_c^m$ to $\mathbb{B}_c^n$ is in turn defined as

$$
\boldsymbol{y} = \mathcal{F}^c(\boldsymbol{x}; Z, \boldsymbol{r}) = \frac{\boldsymbol{w}}{1 + \sqrt{1 + c\|\boldsymbol{w}\|^2}}, \quad \boldsymbol{w} = \left( \frac{1}{\sqrt{c}} \sinh \left( \sqrt{c} v_k(\boldsymbol{x}) \right) \right)_{k=1}^n,
\tag{6}
$$

where the $v_k(\cdot)$ are the scores from the Poincaré multinomial logistic regression and where $Z = [\boldsymbol{z}_1|...|\boldsymbol{z}_n] \in (\mathcal{T}_0 \mathbb{B}_c^m)^n = \mathbb{R}^{m \times n}$ and $\boldsymbol{r} = (r_k)_{k=1}^n \in \mathbb{R}^m$ are the parameters of the layer. Given hyperbolic fully connected layers, Van Spengler et al. provide formulations for 2D convolutions, batch normalization and the ReLU activation in hyperbolic space, along with their weight initialization. We use these blocks as a starting point to develop a fully Hyperbolic U-Net.

## 3.1. Hyperbolic U-Net

We consider the problem of image segmentation where we are given an input image $\boldsymbol{x} \in \mathbb{R}^{H \times W \times 3}$, with height $H$ and width $W$ of the image, respectively. For each pixel $\boldsymbol{x}_{ij} \in \mathbb{R}^3$, $i = 1, ..., H$, $j = 1, ..., W$, we need to assign a label $\boldsymbol{y}_{ij} \in \{1, ..., C\}$, denoting one of $C$ classes. Let $f(\boldsymbol{x}) : \mathbb{R}^{H \times W \times 3} \to \mathbb{R}^{H \times W \times C}$ denote the function that transforms each pixel to a probability distribution over all $C$ classes per pixel. The U-Net architecture is highly effective at approximating this function (Ronneberger et al., 2015; Isensee et al., 2021). Therefore, we strive to formulate a geometric equivalent of the U-Net architecture in the Poincaré ball model, which we name Hyperbolic U-Net.

U-Net typically consists of four encoder and four decoder blocks with skip connections from the encoder to the decoder. The encoder blocks comprise convolution layers, batch normalizations, ReLU activations, and max pooling layers. The decoder blocks are made of identical layers that inverse these operations, where max pooling is replaced either with a transposed convolution or bilinear upsampling. To create a fully hyperbolic U-Net, all these operations need to be reformulated in hyperbolic space. Below, we outline how to formalize and construct (i) Poincaré 2D transposed convolutions and (ii) hyperbolic bilinear upsampling, and (iii) how to effectively initialize hyperbolic convolutional neural networks.

To embed Euclidean pixel vectors into hyperbolic space, we use the exponential map at the origin, $\exp_{\mathbf{0}}^c : \mathcal{T}_{\mathbf{0}}\mathbb{B}_c^n \to \mathbb{B}_c^n$. Thus, each pixel is mapped as $\hat{\boldsymbol{x}}_{ij} = \exp_{\mathbf{0}}^c(\boldsymbol{x}_{ij}) \in \mathbb{B}_c^3$. Consequently, let our hyperbolic network produce, for each pixel at location $(i, j)$, an output $\hat{\boldsymbol{z}}_{ij} \in \mathbb{B}_c^C$ which lies in the Poincaré ball. To convert these hyperbolic outputs into Euclidean logits, we apply the logarithmic map at the origin $\log_{\mathbf{0}}^c : \mathbb{B}_c^n \to \mathcal{T}_{\mathbf{0}}\mathbb{B}_c^n$. Using this, the hyperbolic pixel-wise outputs are mapped to Euclidean logits $\boldsymbol{l}_{ij} = \log_{\mathbf{0}}^c(\hat{\boldsymbol{z}}_{ij}) \in \mathbb{R}^C$.

### 3.2. Poincaré Transposed Convolutions

Image-to-image networks require upscaling. Here, we formalize the 2D transposed convolution operation in the Poincaré ball model by extending the geometric principles of the Poincaré convolution layer. Let the input image $\boldsymbol{x}$ have pixel values $\boldsymbol{x}_{kl} \in \mathbb{B}_c^{C_{in}}$, $k = 1, ..., H_{in}$, $l = 1, ..., W_{in}$, where $C_{in}$ is the number of input channels and where $H_{in}$ and $W_{in}$ are the height and width of the image, respectively. Then we can define a 2D Poincaré transposed convolution operation with $C_{out}$ output channels with pixel values $\boldsymbol{h}_{ij} \in \mathbb{B}_c^{C_{out}}$, $i = 1, ..., H_{out}$, $j = 1, ..., W_{out}$. For each input pixel $\boldsymbol{x}_{kl}$, we compute an output receptive field of size $K \times K$, with $K$ odd, that determines the output pixels $\boldsymbol{h}_{ij}$ where $k - \lfloor \frac{K}{2} \rfloor \leq i \leq k + \lfloor \frac{K}{2} \rfloor$, $l - \lfloor \frac{K}{2} \rfloor \leq j \leq l + \lfloor \frac{K}{2} \rfloor$. We denote the output receptive field centered at $(k, l)$ by $Y_{kl}$. Analogous to the Euclidean transposed convolution, the output values of this receptive field are computed by applying a Poincaré fully connected layer $\mathcal{F}^c$ with parameters $Z$ and $r$, as defined in equation 6, to the input pixels and then splitting the output into $K^2$ individual vectors in $\mathbb{B}_c^{C_{out}}$, so $\boldsymbol{Y}_{kl} = \mathcal{S}_{K^2 C_{out} \to C_{out}}(\mathcal{F}^c(x_{kl}; Z, r))$. Note that similar to the Poincaré convolution operation introduced by Van Spengler et al., usual splitting is inappropriate for vectors on the Poincaré ball, as this can result in vectors outside the manifold. Therefore, we employ the $\beta$-split operation defined by Shimizu et al. as follows:

$$\mathcal{S}_{K^2 C_{out} \to C_{out}}(\mathbf{z}) = \left( \exp_{\mathbf{0}}^c \left( \beta_{C_{out}} \beta_{K^2 C_{out}}^{-1} \mathbf{v}_i \right) \right)_{i=1}^{K^2}, \quad (\mathbf{v}_1^T, \ldots, \mathbf{v}_{K^2}^T)^T = \log_{\mathbf{0}}^c(\mathbf{z}), \qquad (7)$$

where $\beta_i = B(\frac{n}{2}, \frac{1}{2})$ with $B$ being the beta function. This splitting operation takes as input a single hyperbolic vector in $\mathbb{B}_c^{K^2 C_{out}}$ and splits it into $K^2$ vectors in $\mathbb{B}_c^{C_{out}}$ such that the average of the Poincaré norms of the output vectors is equal to the Poincaré norm of the input.

### 3.3. Hyperbolic Bilinear Upsampling

While transposed convolutions can be employed to learn task-specific upsampling filters, they can also be compute intensive and increase the number of learnable parameters. This

is often mitigated in U-Net by replacing them with bilinear upsampling. Therefore, we introduce the hyperbolic analogue of bilinear upsampling for images in the Poincaré ball model. Euclidean bilinear upsampling is performed using linear interpolation first in one direction, and then again in another direction. In the hyperbolic setting, we retain the same grid structure and interpolation weights, but we replace all Euclidean midpoint computations with their geodesic counterparts in the Poincaré ball.

For an upsampling factor $s$ and input size $(W_{in}, H_{in})$, each output coordinate $(i, j)$ corresponds to a fractional location $(u, v)$ in the input image, where $u = \frac{i}{s}, v = \frac{j}{s}$. Let $(k, l), (k+1, l), (k, l+1)$, and $(k+1, l+1)$ be the four neighboring input pixels surrounding $(u, v)$, and let the Euclidean interpolation weights be $\alpha = u - k$, $\beta = v - l$. In the Poincaré ball model, we perform a repeated geodesic interpolation, where each pairwise midpoint is replaced by the geodesic midpoint defined by the exponential and the logarithmic maps (see equation 3) at the relevant tangent space. For two hyperbolic vectors $\boldsymbol{a}, \boldsymbol{b} \in \mathbb{B}_c^d$, their weighted geodesic interpolation with weight $t \in [0, 1]$ is $\gamma(\boldsymbol{a}, \boldsymbol{b}; t) = \exp_{\boldsymbol{a}}^c(t \log_{\boldsymbol{a}}^c(\boldsymbol{b}))$. Using the operator $\gamma$, the hyperbolic bilinear upsampling produces each output pixel as

$$\boldsymbol{h}_{ij} = \gamma\Big(\gamma\big(\boldsymbol{x}_{k,l}, \boldsymbol{x}_{k+1,l}; \alpha\big), \gamma\big(\boldsymbol{x}_{k,l+1}, \boldsymbol{x}_{k+1,l+1}; \alpha\big); \beta\Big), \tag{8}$$

which lies in $\mathbb{B}_c^{C_{in}}$ by construction and $i = 1, .., sH_{in}, j = 1, ..., sW_{in}$.

### 3.4. Newton-Scaled Weight Initialization

Shimizu et al. propose an initialization for the Poincaré fully connected and convolutional layers, that produces expressive features but doesn't ensure norm preservation. The identity initialization proposed by Van Spengler et al. solves this for $m \leq n$ (input and output dimensions). However, U-Net comprises encoder and decoder blocks, and for the decoder, the number of input features exceeds the number of output features. In these layers, both initializations fail to preserve the norm, leading to vanishing or exploding hyperbolic norms. Moreover, identity initialization provides too little feature diversity, which can slow convergence and degrade performance.

We introduce a Newton-scaled hyperbolic weight initialization. We first apply a standard Euclidean initialization (e.g., Kaiming or orthogonal) to the Poincaré fully connected layers in hyperbolic convolution or transposed convolution operations, and subsequently rescale each weight matrix by a scalar $s > 0$ to enforce hyperbolic norm preservation. For the Poincaré fully connected layer $\boldsymbol{y} = \mathcal{F}^c(\boldsymbol{x}; Z, \boldsymbol{r})$ mapping inputs $\boldsymbol{x} \in \mathbb{B}_c^m$ to $\boldsymbol{y} \in \mathbb{B}_c^n$, we seek a scalar $s$ such that $\mathbb{E}_{\boldsymbol{x}}\Big[d_c\big(\mathcal{F}^c(\boldsymbol{x}; sZ, \boldsymbol{r}), 0\big)^2\Big] \approx \mathbb{E}_{\boldsymbol{x}}\Big[d_c(\boldsymbol{x}, 0)^2\Big]$, ensuring that the average squared hyperbolic distance (Equation 4) to the origin is preserved across layers. We define

$$g(s) = \mathbb{E}_{\boldsymbol{x}}\Big[d_c(\mathcal{F}^c(\boldsymbol{x}; sZ, \boldsymbol{r}), 0)^2\Big] - \mathbb{E}_{\boldsymbol{x}}\Big[d_c(\boldsymbol{x}, 0)^2\Big], \tag{9}$$

where in practice both expectations are approximated using a randomly sampled batch of training data, and we approximate the root of $g(s)$ using Newton's method. The derivative $g'(s)$ is obtained via automatic differentiation, enabling an efficient and stable layer-wise optimization procedure. Given a randomly sampled batch, inputs to each layer are collected during a single forward pass. Then, for each individual layer, Newton iterations are

performed in closed form, and the final scalar $s$ is applied directly to the weight matrix $Z$ of the layer. The entire process then becomes: (i) each hyperbolic (transposed) convolutional layer is initialized using either Kaiming or orthogonal initialization applied to its Euclidean parameter $Z$; (ii) during a single forward pass with a randomly sampled batch of inputs, we record the features passed to each hyperbolic layer; and (iii) for each layer we compute the scalar $s$ solving $g(s) = 0$ using 5 to 10 Newton iterations, and update $Z \leftarrow sZ$.

This procedure produces an initialization that is as expressive as Kaiming or orthogonal initialization while enforcing empirical hyperbolic norm preservation, even when $m > n$, such as in transposed convolutions. In practice, this leads to more stable training in Hyperbolic U-Net, through improved feature diversity, and faster convergence, while avoiding the exploding-norm issues observed with identity initialization in upsampling layers.

## 4. Experiments

We focus on three types of experiments: (i) an exhaustive comparative evaluation on noise robustness; (ii) ablation studies on the transpose and interpolation layers; and (iii), ablation studies on the Newton approximation-scaled weight initialization.

**Datasets.** We evaluate on seven diverse medical imaging datasets spanning different modalities and anatomical regions such as skin lesion (ISIC16 (Codella et al., 2018), ISIC18 (Tschandl et al., 2018)), breast ultrasound (BUSI (Al-Dhabyani et al., 2020)), polyp (KVASIR (Jha et al., 2019), SANET (Wei et al., 2021)), and dental caries lesion (ACTA (Gonzalez-Valenzuela et al., 2025), DCBR (Tichỳ et al., 2023)) datasets. All details are provided in Appendix E.

**Noise simulation.** We inject four types of noise commonly encountered in medical imaging such as Gaussian, Speckle, Poisson, and Rician noise. We also test the robustness to brightness shifts and contrast variations. The complete range of values is provided in Appendix F. Models are trained *only on clean data*, using the original, unmodified images from each dataset, that is, no noise augmentation or synthetic corruption is applied during training. Models are then evaluated on both the original test sets and their noise-corrupted counterparts, allowing us to measure out-of-the-box robustness without any noise-specific adaptations. The simulated perturbations are chosen to reflect distortions that arise naturally in clinical acquisition pipelines, such as thermal and electronic noise (Gaussian), ultrasound artifacts (Speckle), photon-counting noise in low-dose settings (Poisson), and MRI background noise (Rician), making the evaluation clinically grounded rather than purely synthetic.

**Implementation details.** We investigate multiple U-Net variants, including U-Net++ (Zhou et al., 2019a) and a lightweight U-Net (Ronneberger et al., 2015) architecture with 4 levels and initial feature dimension of 8. All hyperbolic models use trainable curvature initialized to $c = 0.1$. We train with Dice-Focal loss, Riemannian Adam (Bécigneul and Ganea, 2019) optimizer ($lr = 10^{-3}$, weight decay $= 10^{-4}$), batch size 8, for 50 epochs. We deliberately avoid data augmentation to evaluate inherent geometric robustness. In the main paper, we show results for Hyperbolic U-Net with transposed convolution; the results with bilinear interpolation are in Appendix G. We also compare robustness of our models with a lightweight nnU-Net (Isensee et al., 2021, 2024) with the same architecture as our

models. For this, we use the nnU-Netv2 implementation and the details are mentioned in Appendix M.

**Evaluation metrics.** We report Dice score (DSC), mean Intersection over Union (mIoU), dataset IoU (dIoU), Hausdorff Distance (HD) and HD95. We focus on DSC in the main paper, with the other results in Appendix C.

### 4.1. Robustness Through Hyperbolic Geometry

**Overview.** Table 1 presents DSC on clean data and under mid-high noise levels. For all runs, we use transposed convolutions in the decoder. On clean test sets, Hyperbolic U-Net achieves the same average performance as Euclidean U-Net (average DSC: 0.770 vs. 0.765). Under noise, Hyperbolic U-Net significantly outperforms Euclidean U-Net across all datasets and noise types with average improvement of 48% on Gaussian noise, 22% on Speckle noise, 34% on Poisson noise, 42% on Rician noise, 17% on Brightness shift and 17% on Contrast variation. Specifically, we find that adding strong levels of noise hardly affects the DSC for Hyperbolic U-Net, highlighting that a hyperbolic foundation makes U-Net inherently noise robust. In Appendix A and D, we show more results on U-Net++ and nnU-Net architectures, which yield the same conclusions.

**Noise curves and qualitative examples.** Figure 1 shows DSC degradation curves across noise intensities for ISIC16. Hyperbolic U-Net/U-Net++ exhibit hardly any degradation, while the performance of their Euclidean counterparts collapses when noise increases. nnU-Net is more robust to noise than standard U-Net and U-Net++, due to the use of noise in data augmentation. However, even nnU-Net suffers from performance degradation. Our hyperbolic approach, only trained on clean data, obtains the best performance, especially when noise is most severe. Figure 2 provides qualitative comparisons.

**Geometric explanation of noise robustness.** To further investigate why Hyperbolic U-Net is more robust to noise, we analyze the geometric properties of hyperbolic embeddings. Hyperbolic space exhibits exponential volume growth, unlike the polynomial growth in Euclidean space, which leads to larger relative distances between points as a function of their norm. In the context of representation learning, this implies that features corresponding to different semantic classes are more widely separated, even when local variations exist. Consequently, perturbations introduced by noise are less likely to move a representation across class boundaries, effectively increasing the margin to decision boundary overlap.

To validate this explanation, we compute inter-class distances in the learned feature space (Appendix J). We find that hyperbolic embeddings consistently exhibit higher class separation ratios compared to Euclidean embeddings. These results suggest that the observed robustness of Hyperbolic U-Net is rooted in its geometry as the hyperbolic representation space inherently creates more noise-tolerant embeddings, which preserves segmentation accuracy under various noise conditions.

### 4.2. Transpose Convolution vs. Bilinear Upsampling in Hyperbolic Space

This paper outlines two ways to perform upscaling: hyperbolic transposed convolution vs. hyperbolic bilinear upsampling. In Appendix G we show that both achieve similar DSC on clean data as well as under noise. Bilinear upsampling however, reduces parameters by 57%, while the memory consumption ($\approx 1.4$ GB) and training time remain identical. We

| Dataset | Model | Clean | Gaussian | Speckle | Poisson | Rician | Brightness | Contrast |
|---------|-------|-------|----------|---------|---------|--------|------------|----------|
| ISIC16 | U-Net | *0.92* | 0.52 | 0.75 | 0.62 | 0.72 | 0.79 | 0.75 |
| | Hyp U-Net | *0.91* | **0.90** | **0.90** | **0.90** | **0.90** | **0.86** | **0.87** |
| ISIC18 | U-Net | *0.89* | 0.54 | 0.55 | 0.59 | 0.46 | 0.79 | 0.73 |
| | Hyp U-Net | *0.87* | **0.86** | **0.86** | **0.86** | **0.86** | **0.84** | **0.79** |
| BUSI | U-Net | *0.82* | 0.41 | 0.63 | 0.55 | 0.41 | 0.57 | 0.56 |
| | Hyp U-Net | *0.80* | **0.79** | **0.79** | **0.80** | **0.79** | **0.79** | **0.76** |
| SANET | U-Net | *0.78* | 0.49 | 0.56 | 0.49 | 0.50 | 0.52 | 0.61 |
| | Hyp U-Net | *0.76* | **0.67** | **0.71** | **0.69** | **0.67** | **0.67** | **0.67** |
| KVASIR | U-Net | *0.86* | 0.48 | 0.65 | 0.50 | 0.55 | 0.64 | 0.55 |
| | Hyp U-Net | *0.83* | **0.70** | **0.71** | **0.69** | **0.73** | **0.70** | **0.76** |
| ACTA | U-Net | *0.50* | 0.50 | 0.50 | 0.50 | 0.50 | 0.50 | 0.50 |
| | Hyp U-Net | *0.54* | **0.54** | **0.54** | **0.54** | **0.53** | **0.53** | **0.52** |
| DCBR | U-Net | *0.62* | 0.51 | 0.54 | 0.51 | 0.51 | 0.48 | 0.53 |
| | Hyp U-Net | *0.65* | **0.63** | **0.60** | **0.60** | **0.58** | **0.58** | **0.58** |

Table 1: **Robust medical image segmentation Dice scores.** We report the effect of hyperbolic U-Net versus a standard U-Net, with the same architecture and number of parameters, on seven datasets and six noise types. We use the following settings for all: Gaussian ($\sigma_g = 0.2$), Speckle ($\sigma_s = 0.3$), Poisson ($\lambda = 10$), Rician ($\sigma_r = 0.2$), Brightness ($\Delta_b = 0.5$), Contrast ($\Delta_c = 0.3$). We find that a hyperbolic U-Net is much more robust to noise, brightness and contrast shifts.

hypothesize that this may be due to the fact GPUs we use are currently not optimized for hyperbolic operations. We conclude that both approaches are viable as decoders, with bilinear upsampling preferable in resource-constrained clinical settings.

Importantly, while both approaches are viable for standard U-Net decoders, we observe a practical difference when extending to nested decoder architectures such as U-Net++. In this setting, the hyperbolic transposed convolution variant could not be trained due to GPU memory exhaustion, whereas the bilinear upsampling variant remained trainable. We attribute this to the higher number of hyperbolic operations required by transposed convolution, which are repeatedly invoked in the nested computational graph of U-Net++, while bilinear upsampling is a single-shot operation. We conclude that both approaches are viable as decoders, with bilinear upsampling preferable in resource-constrained clinical settings.

### 4.3. Effects of Newton-Approximation Weight Initialization

To validate our initialization, we compare against the identity initialization from Van Spengler et al. and normal distribution initialization from Shimizu et al. for two analyses:

**Feature diversity.** Figure 3 (left) visualizes feature maps from random Gaussian blob inputs after hyperbolic transposed convolution layer (input = 2D, output = 4D). Our initialization produces diverse features across channels, while identity initialization yields either near-zero or redundant feature maps, indicating poor gradient flow. **Norm preservation.** Figure 3 (right) shows output-to-input norm ratios across different input-to-output feature ratios. We start at input features = 16 and vary the output features to

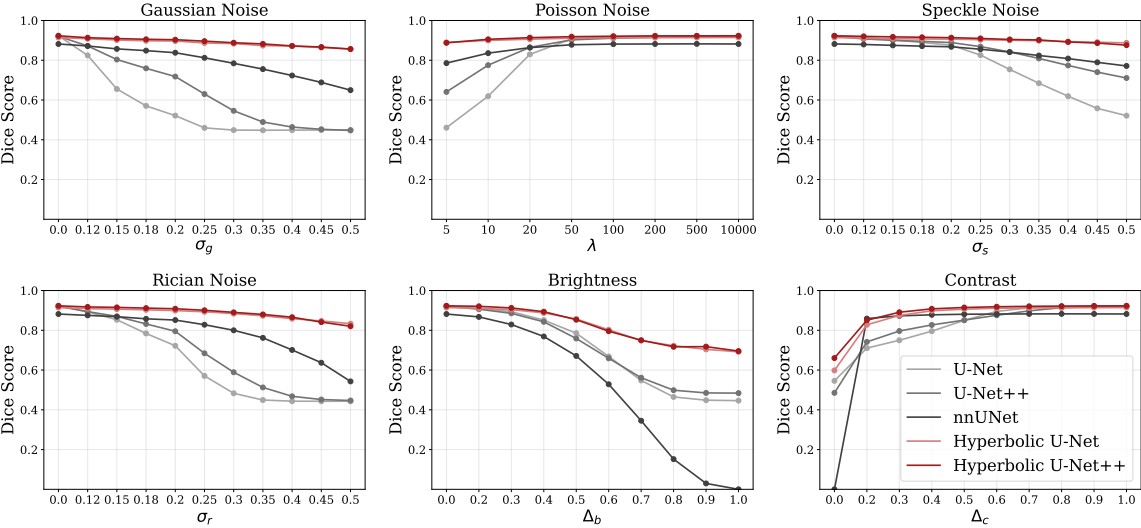

Figure 1: **Performance curves (DSC) on ISIC16 for all noise types.** Hyperbolic U-Net and U-Net++ can handle strong noise interferences, outperforming their Euclidean counterparts and nnUNet.

get different input-to-output feature ratios. Only our initialization maintains stable norms, making our approach the most desirable way to initialize a Hyperbolic U-Net.

## 5. Conclusions

In this paper, we propose Hyperbolic U-Net, the first U-Net architecture fully in hyperbolic space. We introduce hyperbolic formulations of three core Euclidean U-Net components. First, we formulate a geometric equivalent of transposed convolutions in hyperbolic space. Second, we generalize bilinear upsampling through geodesic interpolation. Third, we introduce a new weight initialization method that approximately preserves hyperbolic norms across layers, irrespective of input and output dimensions. Across extensive experiments spanning seven datasets, six noise types, three architectures, and five evaluation metrics, we find that Hyperbolic U-Net consistently demonstrates improved robustness to noise, supporting our hypothesis that hyperbolic geometry can provide more noise-tolerant embeddings. Analysis of the learned feature space suggests that the geometric properties of hyperbolic space, resulting in increased separation between classes, likely contribute to this robustness. Our results highlight the strong potential of hyperbolic geometry for medical image segmentation. A promising future direction is the design of hybrid Euclidean–hyperbolic architectures where early Euclidean layers can extract local features efficiently, while deeper hyperbolic layers enhance robustness to noise.

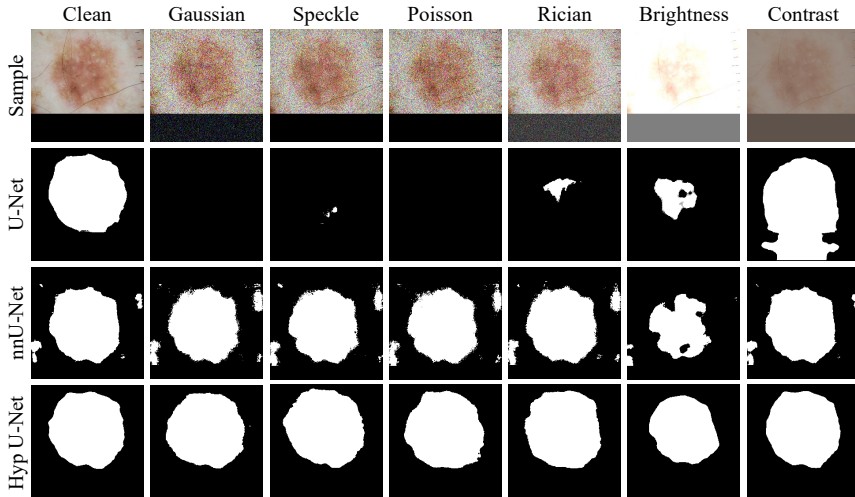

Figure 2: **Qualitative example** of Hyperbolic versus Euclidean U-Net on a skin lesion image for all noise types. We report the predictions on a sample from each dataset on the following perturbations: Gaussian ($\sigma_g = 0.2$), Speckle ($\sigma_s = 0.3$), Poisson ($\lambda = 10$), Rician ($\sigma_r = 0.2$), Brightness ($\Delta_b = 0.5$), Contrast ($\Delta_c = 0.3$). Segmentation predictions of Hyperbolic U-Net remain relatively stable under mid-high levels of noise compared to Euclidean U-Net.

We provide more examples in Appendix B.

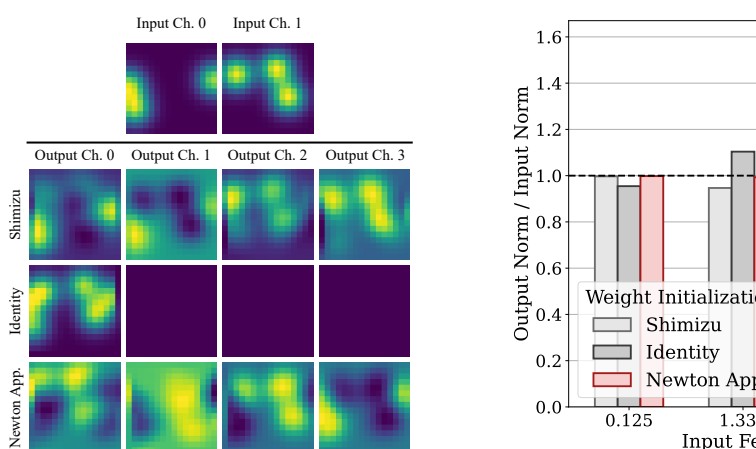

Figure 3: **Effect of our initialization.** Left: we visualize the inputs and the output feature maps obtained from transposed convolution layer. Shimizu et al. and our approach produces expressive features. Right: we show the output-to-input ratios after hyperbolic transposed convolutions for various feature ratios. Only our approach retains the desired ratio of 1.

## Acknowledgments

Swasti S. Mishra acknowledges support from the University of Amsterdam Data Science Centre, as part of the Human Aligned Video AI Lab. Max van Spengler acknowledges the University of Amsterdam Data Science Centre for financial support.

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

## Appendix A. Robustness Evaluation: Noise Curves

Figure A1 shows DSC degradation curves across noise intensities for BUSI dataset. Hyperbolic U-Net/U-Net++ exhibit almost no degradation even at high noise levels, while the performance of Euclidean U-Net/U-Net++ degrades quickly. Although, the performance of nnU-Net stays relatively stable for Poisson and Speckle noise, it degrades for other noises and brightness and contrast shifts.

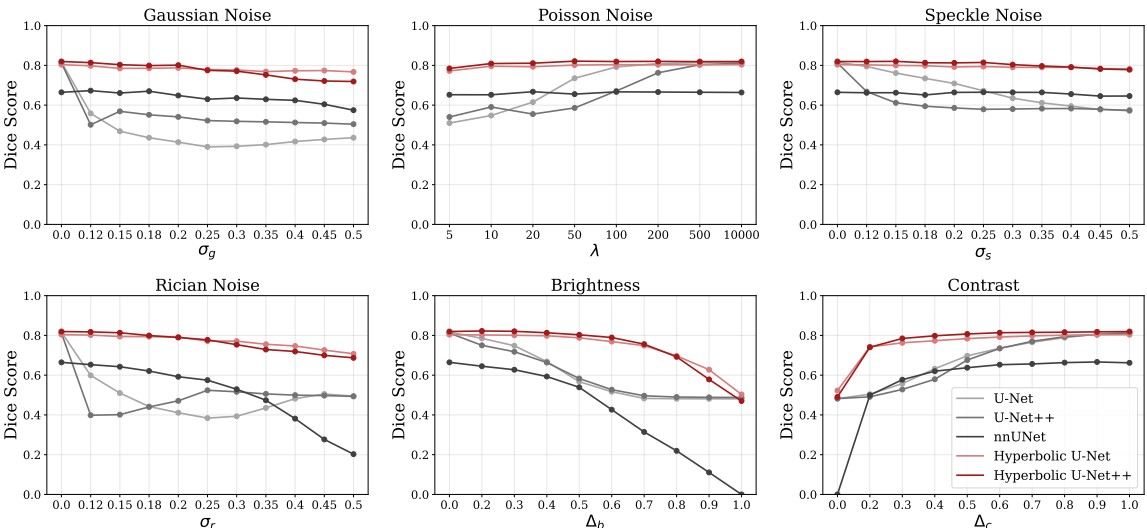

Figure A1: DSC degradation curves across noise intensities for BUSI dataset.

Figure A2 shows DSC degradation curves across noise intensities for KVASIR dataset. Hyperbolic U-Net/U-Net++ and nnU-Net show relatively less degradation overall compared to Euclidean U-Net/U-Net++, which degrades quickly. Hyperbolic U-Net/U-Net++ and nnU-Net perform similarly across Gaussian, Poisson and Speckle noise. However, Hyperbolic U-Net/U-Net++ remains robust to all perturbations.

## Appendix B. Robustness Evaluation: Qualitative Results

Figure B1 and B2 shows qualitative comparisons of Hyperbolic U-Net and Euclidean U-Net on BUSI and KVASIR datasets, respectively. We report the predictions on a sample from each dataset on the following perturbations: Gaussian ($\sigma_g = 0.2$), Speckle ($\sigma_s = 0.3$), Poisson ($\lambda = 10$), Rician ($\sigma_r = 0.2$), Brightness ($\Delta_b = 0.5$), Contrast ($\Delta_c = 0.3$). Segmentation predictions of Hyperbolic U-Net remains relatively stable under mid-high levels of noise compared to Euclidean U-Net.

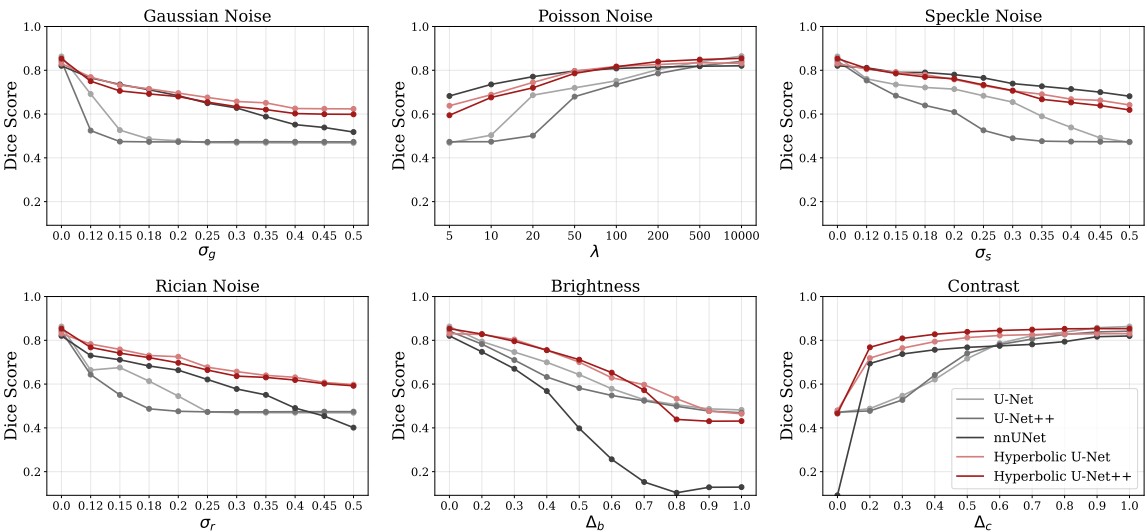

Figure A2: DSC degradation curves across noise intensities for KVASIR dataset.

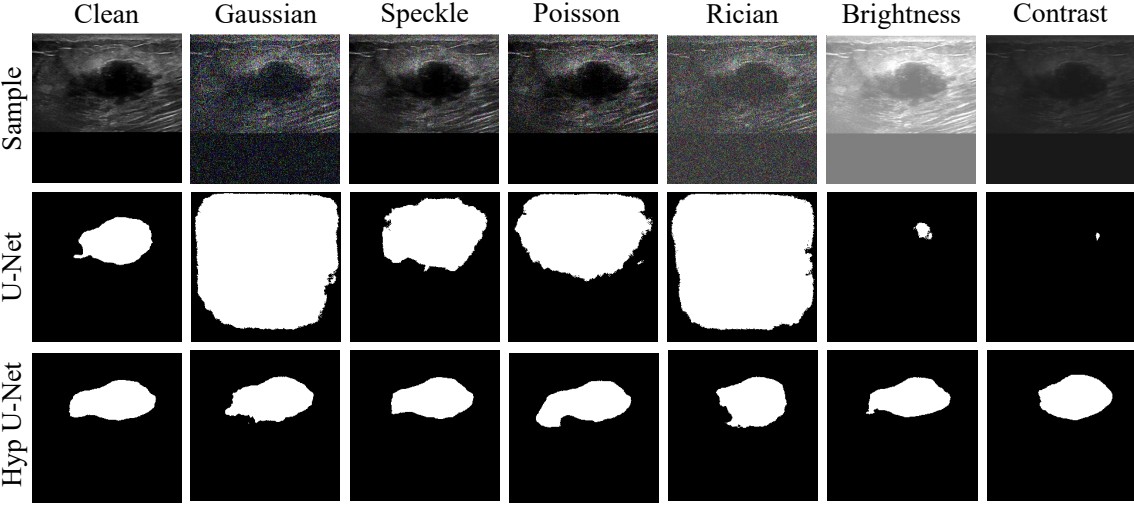

Figure B1: **Qualitative example of Hyperbolic versus Euclidean U-Net on a breast ultrasound image for mid-high level noise types.** We report the predictions on an BUSI dataset sample on the following perturbations: Gaussian ($\sigma_g = 0.2$), Speckle ($\sigma_s = 0.3$), Poisson ($\lambda = 10$), Rician ($\sigma_r = 0.2$), Brightness ($\Delta_b = 0.5$), Contrast ($\Delta_c = 0.3$).

Figure B3 shows qualitative comparisons of Hyperbolic U-Net and nnU-Net under high levels of noise and perturbations. Despite being trained with multiple data augmentation transformations, nnU-Net is not as robust as Hyperbolic U-Net.

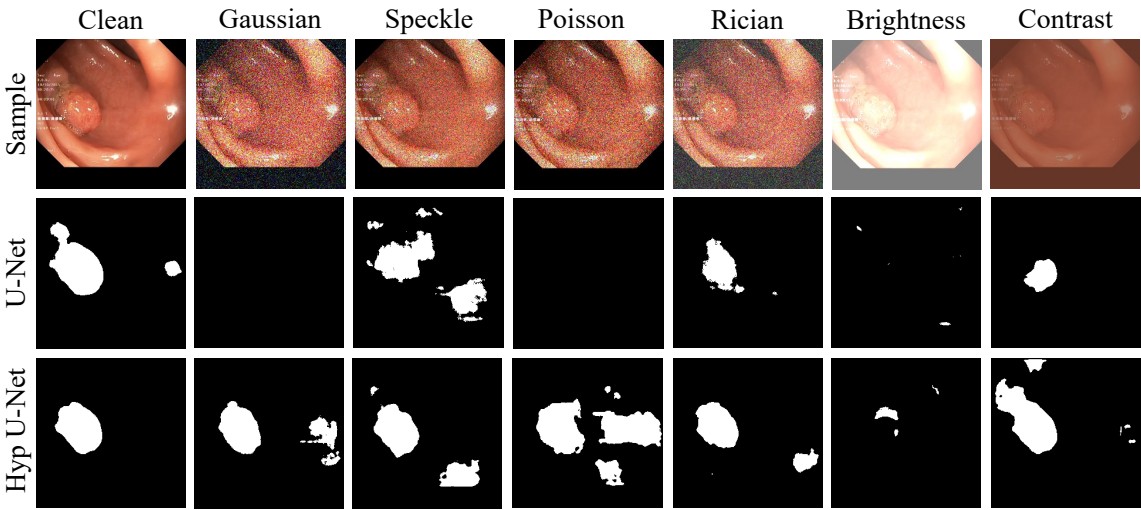

Figure B2: **Qualitative example of Hyperbolic versus Euclidean U-Net for mid-high level noise types.** We report the predictions on an KVASIR dataset sample on the following perturbations: Gaussian ($\sigma_g = 0.2$), Speckle ($\sigma_s = 0.3$), Poisson ($\lambda = 10$), Rician ($\sigma_r = 0.2$), Brightness ($\Delta_b = 0.5$), Contrast ($\Delta_c = 0.3$).

## Appendix C. Evaluation: Hyperbolic U-Net

Table C1 summarizes the evaluation of Hyperbolic U-Net and U-Net on Dice score (DSC), mean Intersection over Union (mIoU), dataset IoU (dIoU), Hausdorff Distance (HD) and HD 95-percentile (95). We report the evaluation metrics of Hyperbolic U-Net versus a standard U-Net, with the same architecture and number of parameters on seven datasets. Both the networks achieve similar scores across most of the metrics.

## Appendix D. Evaluation: Robustness Dice Scores

Table D1 presents the DSC of Hyperbolic U-Net++ versus U-Net++ on clean data and under mid-high noise levels. The results are identical to Table 1. Under noise, Hyperbolic U-Net++ significantly outperforms Euclidean U-Net++ across all datasets and noise types with average improvement of 32% on Gaussian noise, 27% on Speckle noise, 26% on Poisson noise, 37% on Rician noise, 15% on Brightness shift and 21% on Contrast variation.

## Appendix E. Experimental Setup: Datasets

We evaluate on seven diverse medical imaging datasets spanning different modalities and anatomical regions:

- **Skin lesion:** ISIC 2016 (900 train, 379 test) and ISIC 2018 (2594 train, 1000 test) images.

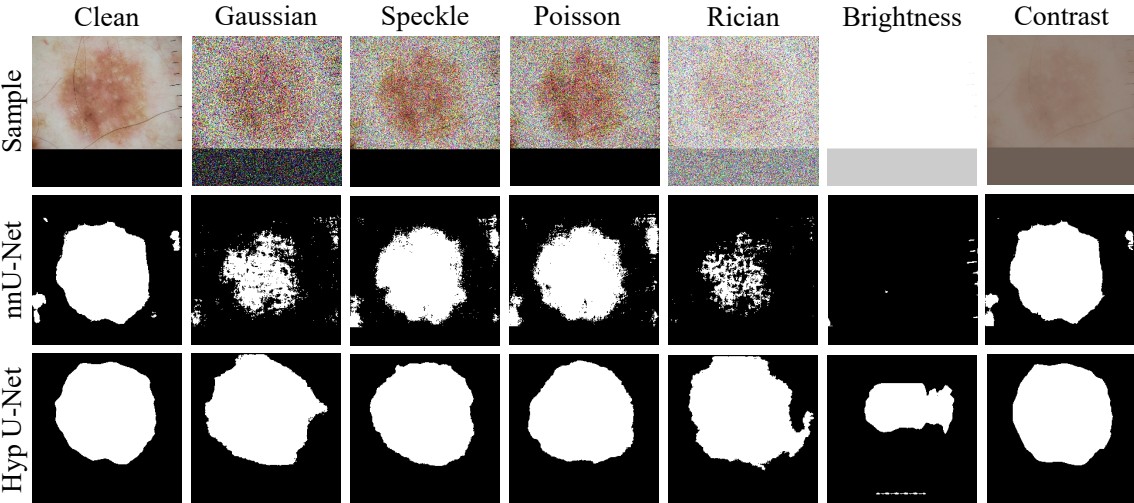

Figure B3: **Qualitative example of Hyperbolic U-Net versus nnU-Net on a skin lesion image for heavy noise types.** We report the predictions on an ISIC16 dataset sample on the following perturbations: Gaussian ($\sigma_g = 0.5$), Speckle ($\sigma_s = 0.5$), Poisson ($\lambda = 5$), Rician ($\sigma_r = 0.5$), Brightness ($\Delta_b = 0.8$), Contrast ($\Delta_c = 0.2$).

- **Breast ultrasound:** BUSI (702 train, 78 test) images.

- **Polyp:** KVASIR (900 train, 100 test) and SANET (1450 train, 798 test) images.

- **Dental caries:** ACTA (2043 train, 227 test) and DCBR (1142 train, 127 test) images.

We use the official train/test splits where available and set aside 10% from the training split for testing in other cases. We also create a validation split during training where we randomly set aside 10% of the data and treat is as a held out set.

**Preprocessing.** We first zero-pad the images wherever the height and the width of the image differ. Then, we resize the images to a $256 \times 256$ resolution in order to reduce the computational costs (crucial in the case of hyperbolic U-Net). All RGB images are then rescaled to lie between $[0, 1]$. For the grayscale images, we use a z-score normalization similar to the one performed in nnU-Net (Isensee et al., 2021).

## Appendix F. Experimental Setup: Test Perturbations

To evaluate robustness under realistic acquisition imperfections, we apply a set of controlled perturbations representing common noise processes and intensity variations encountered in medical imaging. All perturbations are applied independently at test time and swept across multiple severity levels.

| Dataset | Model | DSC↑ | mIoU↑ | dIoU↑ | HD↓ | HD95↓ |
|---|---|---|---|---|---|---|
| ISIC16 | U-Net | **0.92** | 0.91 | 0.91 | 1.43 | 0.09 |
| | Hyp U-Net | 0.91 | **0.93** | **0.93** | **1.18** | **0.06** |
| ISIC18 | U-Net | **0.89** | **0.91** | **0.90** | **0.36** | **0.04** |
| | Hyp U-Net | 0.87 | **0.91** | **0.90** | 0.47 | 0.05 |
| BUSI | U-Net | **0.82** | 0.88 | 0.87 | 13.89 | 4.74 |
| | Hyp U-Net | 0.80 | **0.89** | **0.88** | **9.56** | **3.30** |
| SANET | U-Net | **0.78** | 0.88 | 0.86 | 1.30 | 0.49 |
| | Hyp U-Net | 0.76 | **0.89** | **0.87** | **0.87** | **0.27** |
| KVASIR | U-Net | **0.86** | **0.93** | **0.93** | 9.91 | **0.42** |
| | Hyp U-Net | 0.83 | **0.93** | **0.93** | **9.88** | 0.50 |
| ACTA | U-Net | 0.50 | **0.98** | **0.98** | 35719.49 | 35719.11 |
| | Hyp U-Net | **0.54** | **0.98** | **0.98** | **10.47** | **9.16** |
| DCBR | U-Net | 0.62 | **0.95** | **0.95** | **10.08** | 6.74 |
| | Hyp U-Net | **0.65** | **0.95** | **0.95** | 12.25 | **4.75** |

Table C1: Evaluation of Hyperbolic U-Net and U-Net on Dice score (DSC), mean Intersection over Union (mIoU), dataset IoU (dIoU), Hausdorff Distance (HD) and HD 95-percentile (95).

### F.1. Noise Models

**Channel-wise Gaussian Noise.** We apply additive white Gaussian noise (AWGN) independently per channel:

$$I'_c = I_c + n_c, \quad n_c \sim \mathcal{N}(0, \sigma_c^2),$$

with channel-wise standard deviations

$$\sigma_g = \left\{ \begin{array}{l} [0.00,\ 0.00,\ 0.00], \\ [0.12,\ 0.15,\ 0.18], \\ [0.15,\ 0.20,\ 0.25], \\ [0.18,\ 0.22,\ 0.28], \\ [0.20,\ 0.25,\ 0.30], \\ [0.25,\ 0.30,\ 0.35], \\ [0.30,\ 0.35,\ 0.40], \\ [0.35,\ 0.40,\ 0.45], \\ [0.40,\ 0.45,\ 0.50], \\ [0.45,\ 0.50,\ 0.55], \\ [0.50,\ 0.55,\ 0.60] \end{array} \right\}.$$

**Poisson Noise.** Photon-counting noise is simulated as

$$I' \sim \text{Poisson}(\lambda I),$$

| Dataset | Model | Clean | Gaussian | Speckle | Poisson | Rician | Brightness | Contrast |
|---------|-------|-------|----------|---------|---------|--------|------------|----------|
| ISIC16 | U-Net++ | *0.92* | 0.72 | 0.84 | 0.78 | 0.80 | 0.76 | 0.80 |
|        | Hyp U-Net++ | *0.92* | **0.90** | **0.90** | **0.91** | **0.91** | **0.85** | **0.89** |
| ISIC18 | U-Net++ | *0.89* | 0.54 | 0.55 | 0.59 | 0.46 | 0.79 | 0.73 |
|        | Hyp U-Net++ | *0.87* | **0.86** | **0.86** | **0.86** | **0.86** | **0.84** | **0.79** |
| BUSI | U-Net++ | *0.81* | 0.54 | 0.58 | 0.59 | 0.47 | 0.58 | 0.53 |
|      | Hyp U-Net++ | *0.82* | **0.80** | **0.80** | **0.81** | **0.79** | **0.80** | **0.78** |
| SANET | U-Net++ | *0.80* | 0.49 | 0.49 | 0.49 | 0.49 | 0.58 | 0.62 |
|       | Hyp U-Net++ | *0.78* | **0.59** | **0.58** | **0.56** | **0.60** | **0.67** | **0.69** |
| KVASIR | U-Net++ | *0.84* | 0.47 | 0.49 | 0.47 | 0.48 | 0.58 | 0.53 |
|        | Hyp U-Net++ | *0.85* | **0.68** | **0.71** | **0.68** | **0.70** | **0.71** | **0.81** |
| ACTA | U-Net++ | *0.51* | 0.50 | 0.50 | 0.50 | 0.50 | 0.50 | 0.50 |
|      | Hyp U-Net++ | *0.55* | **0.51** | **0.54** | **0.53** | **0.52** | **0.52** | **0.52** |
| DCBR | U-Net++ | *0.69* | 0.51 | 0.53 | 0.52 | 0.51 | 0.53 | 0.52 |
|      | Hyp U-Net++ | *0.64* | **0.62** | **0.63** | **0.62** | **0.60** | **0.57** | **0.58** |

Table D1: **Robust medical image segmentation Dice scores.** We report the effect of hyperbolic U-Net++ versus U-Net++, with the same architecture and number of parameters, on seven datasets and six noise types. We use the following settings for all: Gaussian ($\sigma_g = 0.2$), Speckle ($\sigma_s = 0.3$), Poisson ($\lambda = 10$), Rician ($\sigma_r = 0.2$), Brightness ($\Delta_b = 0.5$), Contrast ($\Delta_c = 0.3$). We find that a hyperbolic U-Net++ is much more robust to noise as well as brightness and contrast shifts.

where the peak count parameter is varied over

$$\lambda \in \{5,\ 10,\ 20,\ 50,\ 100,\ 200,\ 500,\ 10000\}.$$

These values span extremely noisy low-dose conditions to near noise-free acquisition.

**Channel-wise Speckle Noise.** Speckle noise is applied multiplicatively per channel:

$$I'_c = I_c \cdot (1 + n_c), \quad n_c \sim \mathcal{N}(0, \sigma_c^2),$$

with the same per-channel variance levels as Gaussian noise:

$$\sigma_s = \left\{ \begin{array}{l} [0.00,\ 0.00,\ 0.00], \\ [0.12,\ 0.15,\ 0.18], \\ [0.15,\ 0.20,\ 0.25], \\ [0.18,\ 0.22,\ 0.28], \\ [0.20,\ 0.25,\ 0.30], \\ [0.25,\ 0.30,\ 0.35], \\ [0.30,\ 0.35,\ 0.40], \\ [0.35,\ 0.40,\ 0.45], \\ [0.40,\ 0.45,\ 0.50], \\ [0.45,\ 0.50,\ 0.55], \\ [0.50,\ 0.55,\ 0.60] \end{array} \right\}.$$

**Rician Noise.** MRI-specific noise is modeled as the magnitude of complex Gaussian components:

$$I' = \sqrt{(I + n_1)^2 + n_2^2}, \quad n_1, n_2 \sim \mathcal{N}(0, \sigma^2),$$

with

$$\sigma_r \in \{0.00, 0.12, 0.15, 0.18, 0.20, 0.25, 0.30, 0.35, 0.40, 0.45, 0.50\}.$$

### F.2. Brightness and Contrast Perturbations

**Brightness Shifts.** Brightness is modified as

$$I' = I + \Delta_b,$$

with shift values

$$\Delta_b \in \{0.0, 0.2, 0.3, 0.4, 0.5, 0.6, 0.7, 0.8, 0.9, 1.0\}.$$

**Contrast Scaling.** Contrast is varied using the affine transformation

$$I' = \Delta_c I + (1 - \Delta_c)\mu_I,$$

where $\mu_I$ is the image mean and

$$\Delta_c \in \{0.0, 0.2, 0.3, 0.4, 0.5, 0.6, 0.7, 0.8, 0.9, 1.0\}.$$

## Appendix G. Transposed Convolution vs. Bilinear Upsampling

Table G1 compares the test Dice scores and memory consumption of the proposed Hyperbolic U-Net when employing either transposed convolution or bilinear upsampling within the decoder blocks. Across all seven datasets, the two variants exhibit nearly identical segmentation performance, indicating that the choice of upsampling operation has minimal impact on accuracy. This suggests that the performance gains of our architecture stem primarily from its hyperbolic representation and not from a specific decoder upsampling strategy.

## Appendix H. Derivation of Hyperbolic Bilinear Upsampling

We derive the proposed hyperbolic bilinear upsampling by drawing a direct analogy to classical Euclidean bilinear interpolation and replacing Euclidean geometric primitives with their Riemannian counterparts.

### H.1. Euclidean Bilinear Interpolation

Let the four corners of a unit square be given by vectors

$$\boldsymbol{x}_{00}, \boldsymbol{x}_{10}, \boldsymbol{x}_{01}, \boldsymbol{x}_{11} \in \mathbb{R}^d,$$

located at Euclidean coordinates $(0,0)$, $(1,0)$, $(0,1)$, and $(1,1)$, respectively. For a point $(s, t) \in [0, 1]^2$, Euclidean bilinear interpolation is defined as

$$\boldsymbol{h}(s,t) = (1 - t)\big[(1 - s)\boldsymbol{x}_{00} + s\boldsymbol{x}_{10}\big] + t\big[(1 - s)\boldsymbol{x}_{01} + s\boldsymbol{x}_{11}\big]. \tag{10}$$

| Dataset | Transposed Convolution | | Bilinear Upsample | |
|---|---|---|---|---|
| | Dice score | Mem. Util. | Dice score | Mem. Util. |
| ISIC16 | 0.91 | 1.39 GB | 0.91 | 1.39 GB |
| ISIC18 | 0.87 | 1.39 GB | 0.88 | 1.39 GB |
| BUSI | 0.80 | 1.40 GB | 0.79 | 1.39 GB |
| SANET | 0.75 | 1.40 GB | 0.75 | 1.39 GB |
| KVASIR | 0.83 | 1.40 GB | 0.84 | 1.40 GB |
| ACTA | 0.74 | 1.40 GB | 0.75 | 1.39 GB |
| DCBR | 0.68 | 1.39 GB | 0.67 | 1.38 GB |

Table G1: Dice score and memory utilization comparison between Hyperbolic U-Net with transposed convolution and bilinear upsampling layer.

Equivalently, this operation can be written as two successive linear interpolations. First, interpolate along the horizontal direction:

$$\boldsymbol{h}_0(s) = (1 - s)\boldsymbol{x}_{00} + s\boldsymbol{x}_{10}, \tag{11}$$

$$\boldsymbol{h}_1(s) = (1 - s)\boldsymbol{x}_{01} + s\boldsymbol{x}_{11}, \tag{12}$$

followed by interpolation along the vertical direction:

$$\boldsymbol{h}(s, t) = (1 - t)\boldsymbol{h}_0(s) + t\boldsymbol{h}_1(s). \tag{13}$$

Geometrically, each linear interpolation corresponds to moving along a straight line segment between two points in $\mathbb{R}^d$.

## H.2. From Straight Lines to Geodesics

In a Riemannian manifold $(\mathcal{M}, \mathfrak{g})$, a smooth path $\gamma$ of minimal length between two points $\boldsymbol{a}$ and $\boldsymbol{b}$ is called a geodesic, and can be seen as the generalization of a straight-line in Euclidean space. The geodesic $\gamma$ connecting $\boldsymbol{a}$ and $\boldsymbol{b}$ can be expressed using the exponential and logarithmic maps as

$$\gamma(\boldsymbol{a}, \boldsymbol{b}; t) = \exp_{\boldsymbol{a}}\left(t \log_{\boldsymbol{a}}(\boldsymbol{b})\right), \quad t \in [0, 1].$$

## H.3. Hyperbolic Bilinear Interpolation

We now extend bilinear interpolation to the Poincaré ball model $\mathbb{B}_c^d$ by replacing each Euclidean linear interpolation with its geodesic counterpart.

Let $\boldsymbol{x}_{00}, \boldsymbol{x}_{10}, \boldsymbol{x}_{01}, \boldsymbol{x}_{11} \in \mathbb{B}_c^d$ be four neighboring representations. For interpolation weights $(s, t) \in [0, 1]^2$, we first interpolate along the horizontal direction using geodesics:

$$\boldsymbol{h}_0(s) = \gamma\left(\boldsymbol{x}_{00}, \boldsymbol{x}_{10}; s\right), \tag{14}$$

$$\boldsymbol{h}_1(s) = \gamma\left(\boldsymbol{x}_{01}, \boldsymbol{x}_{11}; s\right). \tag{15}$$

We then interpolate between these intermediate points along the vertical direction:

$$\boldsymbol{h}(s, t) = \gamma\left(\boldsymbol{h}_0(s), \boldsymbol{h}_1(s); t\right). \tag{16}$$

This construction exactly mirrors Euclidean bilinear interpolation, with straight lines replaced by geodesics and linear interpolation replaced by geodesic interpolation.

### H.4. Consistency and Validity

The proposed hyperbolic bilinear interpolation satisfies the following properties:

- **Manifold closure:** Each interpolation step follows a geodesic in $\mathbb{B}_c^d$, ensuring that the output remains within the Poincaré ball.

- **Reduction to the Euclidean case:** As the curvature parameter $c \to 0$, the exponential and logarithmic maps converge to their Euclidean counterparts, and Eq. (16) reduces to classical bilinear interpolation.

- **Geometric consistency:** The construction replaces Euclidean straight-line interpolation with geodesic interpolation, yielding an operation that is consistent with the underlying Riemannian geometry.

Therefore, hyperbolic bilinear upsampling arises as the natural Riemannian generalization of Euclidean bilinear interpolation.

## Appendix I. Derivation of Newton-Scaled Hyperbolic Weight Initialization

We derive the Newton-scaled hyperbolic weight initialization by extending variance-preserving Euclidean initialization schemes to the hyperbolic setting by preserving the expected magnitude of representations measured using the hyperbolic distance.

### I.1. Motivation: Hyperbolic Norm Preservation

In Euclidean neural networks, common initialization schemes such as Kaiming or orthogonal initialization aim to preserve the expected squared $\ell_2$ norm of activations across layers. However, in hyperbolic space, the Euclidean norm is no longer geometrically meaningful. Instead, the notion of feature magnitude is given by the hyperbolic distance to a reference point, typically the origin.

Given a Poincaré fully connected layer $\boldsymbol{y} = \mathcal{F}^c(\boldsymbol{x}; Z, \boldsymbol{r})$, we therefore seek to preserve the expected squared hyperbolic distance $\mathbb{E}_{\boldsymbol{x}}\big[d_c(\boldsymbol{y}, 0)^2\big]$ across layers. This motivates enforcing the condition

$$\mathbb{E}_{\boldsymbol{x}}\big[d_c(\mathcal{F}^c(\boldsymbol{x}; Z, \boldsymbol{r}), 0)^2\big] \approx \mathbb{E}_{\boldsymbol{x}}\big[d_c(\boldsymbol{x}, 0)^2\big]. \tag{17}$$

### I.2. Reduction to a Scalar Scaling Problem

We begin by initializing the Euclidean parameter matrix $Z$ using a standard initialization scheme (e.g., Kaiming or orthogonal). Rather than modifying the structure of the layer, we introduce a scalar rescaling $s > 0$ applied uniformly to $Z$, yielding the scaled mapping

$$\boldsymbol{y}(s) = \mathcal{F}^c(\boldsymbol{x}; sZ, \boldsymbol{r}).$$

This reduces the norm preservation condition in Eq. (17) to finding a scalar root of the function

$$g(s) = \mathbb{E}_{\boldsymbol{x}}\big[d_c(\boldsymbol{y}(s), 0)^2\big] - \mathbb{E}_{\boldsymbol{x}}\big[d_c(\boldsymbol{x}, 0)^2\big]. \tag{18}$$

Importantly, $g(s)$ is a one-dimensional, continuously differentiable function of $s$, since $\mathcal{F}^c$ is smooth with respect to its parameters.

### I.3. Newton's Method for Scalar Root Finding

Since $g(s)$ admits no closed-form root due to the nonlinear nature of hyperbolic operations, we approximate a solution using Newton's method. Starting from an initial guess $s_0 = 1$, the iteration proceeds as

$$s_{t+1} = s_t - \frac{g(s_t)}{g'(s_t)}. \tag{19}$$

The derivative $g'(s)$ is computed efficiently using automatic differentiation, as $\mathcal{F}^c$ is fully differentiable with respect to $s$. In practice, the expectations in Eq. (18) are approximated using a randomly sampled batch of inputs. We found that Newton's method converged rapidly in all encountered configurations.

### I.4. Implementation

The initialization procedure is applied layer-wise, starting from the first layer, as follows:

1. Initialize $Z$ using a standard Euclidean scheme.

2. Perform a single forward pass up to the layer that is to be initialized with a randomly sampled batch to collect layer inputs.

3. Compute $s$ by applying Newton's method to Eq. (18).

4. Rescale the weight matrix as $Z \leftarrow sZ$.

This procedure yields stable initializations even in architectures involving dimensionality expansion, such as transposed convolutions.

## Appendix J. Feature Space Separation Analysis under Noise

To understand the source of the improved robustness observed in Hyperbolic U-Net under increasing noise levels, we analyze the geometric structure of the learned feature representations. In particular, we study how well different semantic classes are separated in the feature space of hyperbolic and Euclidean models, and how this separation evolves as a function of noise degradation.

For both Hyperbolic U-Net and Euclidean U-Net, we extract intermediate feature embeddings from the final decoder stage before the prediction layer. These embeddings correspond to spatial feature vectors associated with each pixel location in the input image.

We quantify feature separation by measuring:

- **Inter-class distances:** distances between feature embeddings corresponding to pixels belonging to different classes (e.g., foreground class vs. background class).

- **Intra-class distances:** distances between feature embeddings corresponding to pixels within the same class.

The distances between feature embeddings are computed using the geometry of the corresponding model, i.e., for the hyperbolic model, pairwise distances are computed using the hyperbolic distance in the Poincaré ball. Whereas, distances for the Euclidean embeddings are computed using the standard Euclidean distance. However, due to the high spatial resolution of the images ($256 \times 256$), we randomly subsample 1024 feature pairs per image to compute these distances. This subsampling strategy is applied consistently across all models, datasets, and noise levels.

### J.1. Separation Ratio

To summarize the relative separation between classes, we define a separation ratio as:

$$\text{Separation Ratio} = \frac{\text{Mean Inter-class Distance}}{\text{Mean Intra-class Distance}}$$

A higher separation ratio indicates stronger relative class separation, corresponding to a larger margin between classes in the feature space.

### J.2. Results

We analyze the separation of learned feature representations for the KVASIR dataset under varying noise conditions similar to our previous robustness experiments. The trends are consistent across different datasets. Figure J1 shows the evolution of inter-class and intra-class distances as a function of noise intensity. Note that, absolute distance values are not directly comparable between models as they are computed in different geometric spaces. The key comparison is the relative separation (shaded area) within each model's geometry.

**Consistent separation advantage.** Across all six perturbation types (Gaussian noise, Poisson noise, speckle noise, Rician noise, brightness, and contrast), Hyperbolic U-Net exhibits larger separation margins (shaded regions) between inter-class and intra-class distances compared to Euclidean U-Net. This larger margin indicates that hyperbolic features maintain better class discriminability in the learned representation space.

To quantify the separation behavior with a single metric, we visualize the mean separation ratio (Figure J2). Higher separation ratios indicate stronger relative class separation. Hyperbolic U-Net consistently achieves separation ratios [18-200%] higher than Euclidean U-Net across all perturbation types, with a mean improvement of 97%.

These results suggest that the hyperbolic geometry provides well-separated feature representations. The inherent exponential growth of volume in hyperbolic space may enable the model to maintain larger margins between semantic classes while keeping within-class features compact, even when the input signal is corrupted. This geometric advantage translates directly into improved segmentation performance under noise, as evidenced by the Dice score improvements reported in the main paper.

## Appendix K. Curvature Ablation Study

To study the effect of curvature on model performance and stability, we conducted an ablation over both fixed and trainable curvature settings in the Poincaré ball model. We

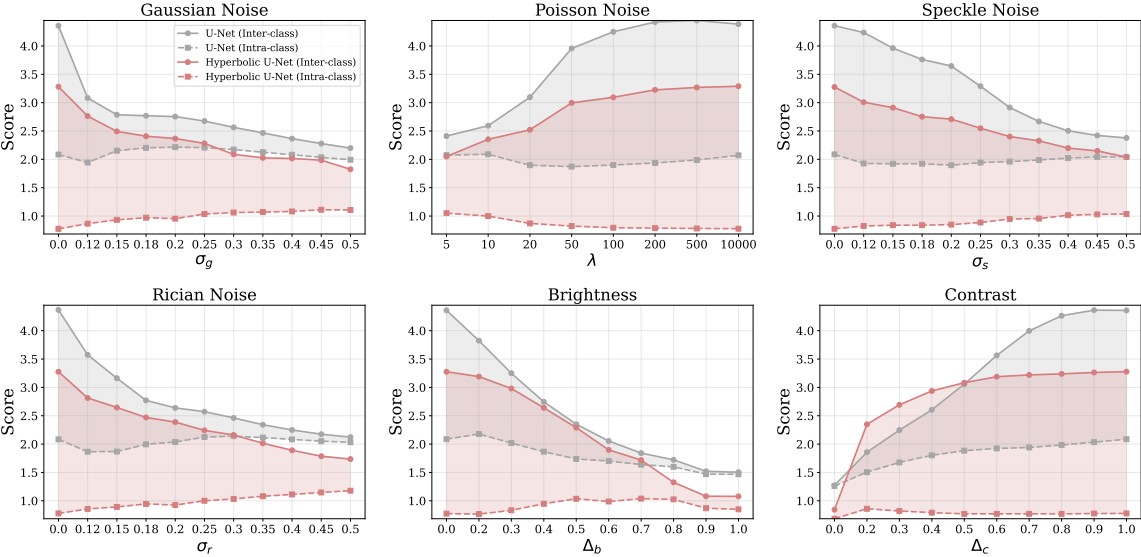

Figure J1: Evolution of feature space separation under increasing noise perturbations. Solid lines represent mean inter-class distances, while dashed lines represent mean intra-class distances for Euclidean U-Net (gray) and Hyperbolic U-Net (red). The shaded regions indicate the separation margin between classes. A larger shaded area corresponds to better class discriminability. Hyperbolic U-Net maintains consistently wider margins across all perturbation types and intensities, indicating superior preservation of geometric structure under noise degradation.

evaluated two commonly used initial curvature values, $c = 0.1$ and $c = 1.0$, consistent with prior work such as Poincaré ResNet (Van Spengler et al., 2023). For all settings, the remaining architecture, optimization parameters, and training protocol were kept identical.

### K.1. Fixed vs. Trainable Curvature

Across all datasets and architectures, we did not observe any strong differences in training stability or segmentation performance between fixed and trainable curvature settings. All models converged reliably without exhibiting numerical instabilities.

When curvature was set to be trainable, initializing at $c = 0.1$ yielded slightly higher Dice scores on average compared to other configurations. Based on this observation, we adopt a trainable curvature initialized at $c = 0.1$ in our main experiments.

### K.2. Learned Curvature Values

When curvature was trainable, the final learned curvature varied depending on several factors, including network depth, initial number of feature channels, dataset, decoder design (transposed convolution vs. bilinear upsampling), and weight initialization. Across all

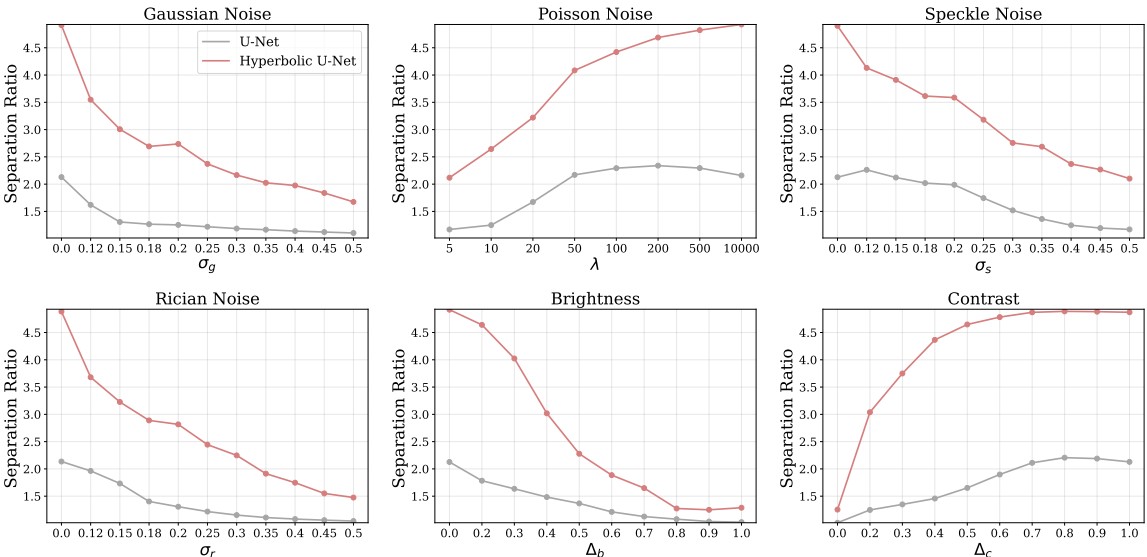

Figure J2: Separation ratio (inter-class distance / intra-class distance) as a function of noise intensity across six perturbation types. Higher values indicate better relative class separation in the learned feature space. Hyperbolic U-Net consistently achieves higher separation ratios than Euclidean U-Net, particularly under severe noise conditions, demonstrating its ability to maintain discriminative feature representations when input quality degrades.

experimental settings, the learned curvature values consistently converged to the range $[0.1, 1.5]$. Importantly, no systematic drift toward extreme curvature values was observed.

### K.3. Performance Sensitivity to Curvature

Overall, segmentation performance was not highly sensitive to the choice of initial curvature. Models initialized with different curvature values achieved comparable Dice scores, and all configurations converged reliably.

This suggests that the Poincaré ball formulation is robust to curvature initialization, particularly when curvature is allowed to be trainable. The ability to adapt curvature during training enables the model to adjust the effective geometry of the representation space to the data and task.

### K.4. Summary

These results indicate that while curvature influences the geometric structure of the embedding space, the proposed Hyperbolic U-Net is not critically dependent on a specific curvature value. A trainable curvature initialized at $c = 0.1$ provides a stable and slightly favorable default, and is therefore used throughout the paper.

## Appendix L. Hyperbolic U-Net Architecture

Here, we detail the architecture of the proposed Hyperbolic U-Net, including its building blocks and the handling of skip connections and norm stability. The architecture is an analogue of the standard Euclidean U-Net, where all operations are replaced by their hyperbolic counterparts while preserving the overall encoder–decoder structure.

### L.1. Overall Architecture

The Hyperbolic U-Net architecture consists of:

- exponential map operation to embed the Euclidean pixel vectors into hyperbolic space

- an encoder composed of repeated Down blocks,

- a decoder composed of repeated Up blocks with skip connections, and

- logarithmic map operation to map the logits back into the Euclidean space.

All feature representations are maintained on the Poincaré ball throughout the network.

### L.2. DoubleConv Block

The fundamental building block of the architecture is the DoubleConv module, which mirrors the Euclidean U-Net design. Each DoubleConv block consists of the following sequence, repeated twice:

(Hyperbolic Convolution $\rightarrow$ Hyperbolic Batch Normalization $\rightarrow$ Hyperbolic ReLU) $\times$ 2

All operations are defined on the Poincaré ball and follow the formulations introduced in Poincaré ResNet (Van Spengler et al., 2023).

### L.3. Down Block (Encoder)

Each Down block in the encoder consists of:

1. Hyperbolic Max Pooling to reduce spatial resolution, followed by

2. a DoubleConv block to increase feature expressivity.

This structure allows the encoder to progressively aggregate global context while remaining entirely within hyperbolic space.

### L.4. Up Block (Decoder)

Each Up block in the decoder performs spatial upsampling followed by feature fusion via skip connections. Two decoder variants are supported:

- Hyperbolic Transposed Convolution, or

- Hyperbolic Bilinear Upsampling

In both cases, upsampling is followed by:

1. Poincaré concatenation between the upsampled decoder features and the corresponding encoder features (**skip connection**), and

2. a DoubleConv block.

The Poincaré concatenation operation, introduced in Hyperbolic Neural Networks++ (Shimizu et al., 2021), ensures that feature fusion preserves the manifold constraints and remains within the Poincaré ball.

### L.5. Summary

The Hyperbolic U-Net is a reformulation of the standard U-Net architecture, where all operations are replaced by their hyperbolic counterparts while preserving the original architectural design. This enables a direct comparison between Euclidean and hyperbolic models while ensuring that all computations respect the underlying manifold structure.

## Appendix M. nnU-Net Baseline Implementation Details

All baseline experiments are conducted using the official nnU-Net v2 implementation, obtained from the publicly released nnU-Net repository maintained by the original authors (Isensee et al., 2021, 2024). This repository corresponds to the nnU-Net v2 framework and includes the updated training, inference, and data preprocessing pipelines.

### M.1. Training and Inference Pipeline

We use the standard nnU-Net v2 training and inference pipelines without modification, including automatic data preprocessing, default loss functions, default optimizer and learning rate schedules, and default data augmentation strategies.

### M.2. Architectural Configuration

To ensure a fair architectural comparison between nnU-Net and our Hyperbolic U-Net models, we align the network depth and initial feature dimensionality across methods. Specifically, the nnU-Net architecture is configured with 4 levels, and an initial feature dimension of 8 channels. All other architectural and hyperparameter settings remain at their default nnU-Net v2 values. No additional tuning or customization is applied.

## Appendix N. Inference-Time Analysis

We evaluate inference-time performance using a batch size of 8 under identical hardware and an input resolution of $(256 \times 256)$. Hyperbolic U-Net incurs a substantially higher inference-time cost than Euclidean U-Net, corresponding to an approximately $11 \times$ increase in batch inference time. This results in a significantly lower throughput in terms of images processed per second. The overhead arises from repeated hyperbolic operations (e.g., exponential and logarithmic maps), which are currently not supported by optimized GPU kernels and are

executed at a higher computational cost. We emphasize that this work focuses on representational robustness rather than inference efficiency, and that improving the efficiency of hyperbolic operations is an important direction for future work.

| Model | Batch Time (ms) $\downarrow$ | Throughput (img/s) $\uparrow$ |
|---|---|---|
| Euclidean U-Net | $1553.34 \pm 112.17$ | 5.15 |
| Hyperbolic U-Net | $18493.52 \pm 332.01$ | 0.43 |

Table N1: Inference-time comparison between Euclidean U-Net and Hyperbolic U-Net using a batch size of 8. Inference time is reported as mean $\pm$ standard deviation over multiple forward passes on the same GPU. Throughput is computed as images processed per second.

## Appendix O. Inference-Time Analysis

We evaluate inference-time performance using a batch size of 8 under identical hardware and an input resolution of $(256 \times 256)$. Hyperbolic U-Net incurs a substantially higher inference-time cost than Euclidean U-Net, corresponding to an approximately $11 \times$ increase in batch inference time. This results in a significantly lower throughput in terms of images processed per second. The overhead arises from repeated hyperbolic operations (e.g., exponential and logarithmic maps), which are currently not supported by optimized GPU kernels and are executed at a higher computational cost. We emphasize that this work focuses on representational robustness rather than inference efficiency, and that improving the efficiency of hyperbolic operations is an important direction for future work.

| Model | Batch Time (ms) $\downarrow$ | Throughput (img/s) $\uparrow$ |
|---|---|---|
| Euclidean U-Net | $1553.34 \pm 112.17$ | 5.15 |
| Hyperbolic U-Net | $18493.52 \pm 332.01$ | 0.43 |

Table O1: Inference-time comparison between Euclidean U-Net and Hyperbolic U-Net using a batch size of 8. Inference time is reported as mean $\pm$ standard deviation over multiple forward passes on the same GPU. Throughput is computed as images processed per second.

## Appendix P. Ablation on Newton-Scaled Weight Initialization

To evaluate the effect of the proposed Newton-scaled weight initialization, we perform an ablation study comparing three initialization strategies: Newton-scaled initialization, Shimizu initialization, and identity initialization, across seven datasets. Table P reports the Dice scores obtained using each initialization under identical training settings.

Newton-scaled initialization consistently achieves comparable or improved Dice scores across most datasets, with more stable optimization during early training as shown in the main paper. These results indicate that the proposed initialization provides a favorable starting point for hyperbolic optimization, though final performance remains largely comparable across initialization schemes.

Our initialization strategy follows the same design principle as standard Euclidean initializations (e.g., Xavier or Kaiming), where weights are constrained only at initialization to ensure stable signal propagation at the start of training. In practice, we observe that subsequent training dynamics, combined with hyperbolic batch normalization are sufficient to prevent uncontrolled norm growth, without requiring explicit norm clamping during training.

| Initialization | ISIC16 | ISIC18 | BUSI | SANET | KVASIR | ACTA | DCBR |
|---|---|---|---|---|---|---|---|
| Identity Init | 0.84 | 0.85 | 0.76 | 0.73 | 0.80 | 0.53 | 0.62 |
| Shimizu Init | 0.89 | 0.87 | 0.78 | 0.74 | 0.82 | 0.54 | 0.63 |
| Newton-Scaled Init | **0.91** | **0.87** | **0.80** | **0.76** | **0.83** | **0.54** | **0.65** |

Table P1: Ablation study on weight initialization strategies. Dice scores are reported for Hyperbolic U-Net trained with different initializations across seven datasets.

