# OpenReview forum: "Hyperbolic U-Net for Robust Medical Image Segmentation"
_MIDL.io/2026/Conference — MIDL 2026 Poster_

### Official Review · Reviewer_3b1q · 2025-12-29

**Confidence:** 4
**Preliminary Rating:** 4
**Final Rating:** 5

**Summary:**

The authors introduce Hyperbolic U-Net, the first fully hyperbolic formulation of the U-Net architecture, by redesigning both encoder and decoder operations, including transposed convolutions and upsampling, within the Poincaré ball model. They further propose a Newton-scaled weight initialization that stabilizes training by preserving hyperbolic norms, enabling deep encoder and decoder architectures to function reliably in hyperbolic space. Across seven medical imaging datasets and multiple realistic noise and intensity perturbations, Hyperbolic U-Net demonstrates substantially improved out-of-the-box robustness compared to Euclidean (standard) U-Net and nnU-Net, without relying on denoising or noise-aware data augmentation.

**Strengths:**

The main strength of this work is its architectural novelty, introducing the first fully hyperbolic U-Net with fully hyperbolic decoder operations rather than limiting hyperbolic geometry to embedding or classification layers. A second major strength is the clear and extensive empirical validation, demonstrating strong out-of-the-box robustness to multiple realistic noise and intensity perturbations across seven medical imaging datasets. Finally, the proposed Newton-scaled initialization seems technically sound as it enables stable training of hyperbolic encoder–decoder networks and could be broadly useful beyond this specific architecture.

**Weaknesses:**

The author provide limited theoretical or geometric intuition, avoiding to explain (or speculate) why hyperbolic geometry yields such strong noise robustness beyond empirical observation. Additionally, Hyperbolic U-Net slightly underperforms Euclidean U-Net on clean data in several datasets, and this robustness–accuracy trade off is not fully discussed. Finally, the computational and inference-time costs of hyperbolic decoding should be further described.

**Detailed Comments:**

-The authors mention a lightweight nnU-Net as comparison method. Was data augmentation avoided also for the nnU-Net? If this is the case, an comparison should be performed using the standard nnU-Net model, as its main strength relies on the data augmentation techniques used.

-Figure 2: please add the noise parameters as in Figure B1.

-Figures with qualitative examples: for a more comprehensive assessment show both euclidean U-Net and nnUNet along with the proposed Hyp U-Net in all figures.

-Figure A2: the legend covers the curves.

-It would be great to add also some failure cases for the Hyperbolic U-Net to better understand its behavior.

**Justification Of Final Rating:**

The authors have carefully addressed all my concerns in details, considerably improving the manuscript. This is a novel work that will be of interest to many researchers in the field. In my opinion, this work deserves to be accepted.

**Justification Of The Preliminary Rating:**

The author introduce a novel hyperbolic segmentation architecture and demonstrate substantial robustness gains across seven medical imaging datasets with a strong evaluation. This work could be of interest to many researchers in the medical imaging field.

**Questions To Address In The Rebuttal:**

-Why the Poincaré ball specifically mitigates random noise so effectively compared to Euclidean space?

-The Hyperbolic U-Net underperforms Euclidean U-Net on clean data in several datasets. This is an important limitation which should be discussed as noise often appears randomly. How do you explain this behavior? Could a hybrid Euclidean–hyperbolic early encoder help?

-At inference, what is the computational cost/time of Hyp U-Net vs Euclidean UNet?

---

> ### Author Response · Authors · 2026-01-25
>
> We thank the reviewer for their insightful comments. We address them below:
>
> **Geometrical motivation for robustess of hyperbolic geometry to noise.**
> We thank the reviewer for this insightful comment and agree that the manuscript benefits from a clearer geometric intuition explaining why hyperbolic geometry, the Poincaré ball model in particular exihibits increased robustness to noise. In the revised manuscript, we add a dedicated discussion clarifying the geometric properties of the Poincaré ball that plausibly contribute to this behavior, together with a supporting experiment (Appendix J).
>
> A key property of the Poincaré ball is its exponential volume growth with respect to geodesic radius, in contrast to the polynomial growth of Euclidean space. This induces an exponential expansion of geodesic distances as representations move away from the origin, enabling different semantic classes to occupy increasingly well-separated regions of the manifold. As a result, random perturbations induced by noise are less likely to move feature representations across class boundaries, compared to Euclidean embeddings where distances grow linearly.
>
> Based on this geometric intuition, we hypothesize that hyperbolic neural networks, such as Hyperbolic U-Net, are inherently more robust to noise than their Euclidean counterparts due to increased relative separation between class representations. We support this hypothesis empirically through extensive experiments across multiple datasets and noise types, and further validate it in Appendix J by measuring inter-class and intra-class distances in the learned feature space. Hyperbolic U-Net consistently achieves higher class separation ratios, providing a geometric explanation for its improved robustness under noise.
>
> **Hyperbolic vs. Euclidean UNet performance.**
> Regarding the performance in the setting without added noise, we find that on average, the mIoU performance is roughly equal (0.91 for Euclidean vs 0.92 for Hyperbolic as per Table C1). We furthermore note Hyperbolic U-Net performs competitively or better on complementary segmentation metrics such as mean Intersection-over-Union (mIoU), dataset IoU (dIoU), Hausdorff Distance (HD), and the 95th percentile Hausdorff Distance (HD95), as shown in Table C1. We conclude that on standard settings, Hyperbolic and Euclidean U-Nets perform roughly on par, while Hyperbolic U-Net is superior in the noisy setting.
>
> **Potential for hybrid architectures.**
> We belive that a hybrid model is a very interesting future direction. Such a design can potentially get the best out of both perspectives. We have included this future direction in the conclusion of the paper.
>
> **Computational and inference time costs.**
> We thank the reviewer for pointing this out. We now report inference-time measurements in Appendix N. Under identical conditions (batch size = 8), Hyperbolic U-Net is approximately 11× slower than Euclidean U-Net at inference, due to the use of hyperbolic operations that are currently not fully optimized.
>
> **Comparison of Hyperbolic UNet and nnU-Net.**
> We thank the reviewer for raising this point. We would like to clarify that all nnU-Net baseline experiments are performed using the standard data augmentation pipeline provided in the official nnU-Netv2 repository, without disabling or modifying any of its default augmentations. This includes the noise-based and intensity-based augmentations that constitute a core strength of nnU-Net. In contrast, Hyperbolic U-Net is trained without any artificial data augmentation. To avoid any ambiguity, we have clarified these implementation details in Section 4 and added an explicit description in Appendix M.
>
> **Textual and figure revisions.**
> We thank the reviewer for these helpful suggestions. The noise parameters in Figure 2 have been added to the figure caption. We have also updated the qualitative results to include Euclidean U-Net, nnU-Net, and Hyperbolic U-Net in Figure 2 and will add them to all comparison figures (B1, B2 and B3), enabling a more comprehensive visual assessment.
>
> In Figure A2, we have reduced the size of the legend to prevent it from occluding the curves. Regarding failure cases, Figure B2 illustrates such an example for Hyperbolic U-Net. In particular, on smaller datasets such as KVASIR, Hyperbolic U-Net exhibits relatively lower Dice scores across both clean and noisy settings.

---

> > ### Comment · Reviewer_3b1q · 2026-01-29
> >
> > Thank you for addressing my concerns, this is a strong work and I have increased my rating.

---

### Official Review · Reviewer_9Q83 · 2026-01-05

**Confidence:** 3
**Preliminary Rating:** 2
**Final Rating:** 3

**Summary:**

The paper proposes an extension to hyperbolic CNNs by proposing transposed convolutions and bilinear up-sampling in  Poincare projection. Additionally, a novel weight initialization method is proposed to preserve the input norms to avoid exploding norms in hyperbolic space due to the exponential map. The proposed hyperbolic Unet is tested on segmentation tasks for multiple medical imaging datasets and modalities.

**Strengths:**

- First implementation of transposed convolutions in Poincare model of hyperbolic spaces
- Proposed hyperbolic Unet perform better when artificial noise introduced to the test data.
- Large number of modalities used in testing

**Weaknesses:**

- The main idea of the paper revolves around robustness to artificial noise. The paper is missing crucial discussion on why hyperbolic spaces, especially Poincare model, are more robust to noise compared to Euclidean space. Even though the experiments indicate the robustness, the main idea is not well grounded.
- There is no ablation for Newton-Scaled Weight Initialization. The claims such as faster convergence need to be supported empirically. Additionally, norm is only preserved at the beginning, but due to the exponential mapping it is necessary to have a constrained norm through the training (via clamping etc.), can the authors explain why it is sufficient to have it constrained only in the initialization?
- The proposed bilinear upsampling does not affect flops, number of parameters nor the performance (Table G1). I am curious why it is proposed as a novelty, even though the main network does not have it.
- The starting curvature is 0.1 which is very flat. There should be a discussion on the curvature, different curvature values if it is fixed, or the last value if it is trainable.
- Missing space in page 3, end of the second paragraph.
- Adam assumes that layer weights live in euclidean space. To the best of my knowledge, Adam's moment updates use vector operations that assume a global coordinate system, this is not the case with hyperbolic spaces where there are tangent spaces that vectors live in. Thus it is necessary to use parallel transport operation when estimating the Adam moments. Riemannian Adam is proposed for this correction [1,2]
- Artificial noise augmentation during training does not change the architecture nor increase the computational cost. From the contrastive learning paradigm, it is known to increase general performance not only the robustness. There should be experiments showing the benefit of hyperbolic Unet over simple noise augmentation
- The baseline comparisons should include more modern architectures such as nnUnetv2 [3].


References:

[1] Bécigneul, Gary, and Octavian-Eugen Ganea. "Riemannian Adaptive Optimization Methods." International Conference on Learning Representations (ICLR 2019)

[2] Nickel, Maximillian, and Douwe Kiela. "Poincaré embeddings for learning hierarchical representations." Advances in neural information processing systems 30 (2017)

[3] Isensee, Fabian, et al. "nnu-net revisited: A call for rigorous validation in 3d medical image segmentation." International Conference on Medical Image Computing and Computer-Assisted Intervention. Cham: Springer Nature Switzerland, 2024.

**Detailed Comments:**

See Weaknesses and Questions to Address in the Rebuttal

**Justification Of Final Rating:**

The authors successfully answered most of my questions, especially related to the implementation wise details such as optimizer, concatenation operation, I believe the paper merits to an acceptance overall

**Justification Of The Preliminary Rating:**

The paper lacks a central hypothesis. Since the experimental focus is on segmentation performance under artificial noise, There should be a hypothesis sentence in the introduction such as "Hyperbolic Unets are more robust to noise than Euclidean Unets, due to the xyz properties of hyperbolic spaces". Otherwise, the paper reads as some experiments are done, and a technical report is written, rather than experiments are designed to confirm the hypothesis. Additionally, there is no experimental evidence for the superiority of the proposed initialization. Lastly, hyperbolic bilinear upsampling is not part of the main model, and does not provide any computational efficiency or performance gain, it shouldnt be listed as novelty.

**Questions To Address In The Rebuttal:**

- Is there a specific reason for Poincare instead of Lorentz or Klein? Lorentz is known to be more numerically stable due to avoiding singularity which is observed when the points are close to the radius of a poincare disk [1]
- Authors should explain which property of hyperbolic spaces induce robustness to noise. In the current manuscript, the paper lacks a hypothesis. It is experimentally "discovered" that hyperbolic spaces are more robust to artificial noise. Current version reads more like a technical report rather than a research paper.
- What is the curvature value after training? How does the curvature affect the performance?
- In the proposed architecture, how are Unet skip connections handled?
- Are there any normalization or clamping function between layers to ensure that norm is under control during training?
- There should be ablation studies on the proposed initialization. Also please clarify why it is sufficient to have the norms constrained only in the beginning of training.
- The function $\gamma$ (used in equation 8) assumes that the interpolated value lives in the tangent space of a. Could authors clarify why it is the case, and a parallel transport is not needed?
- Could authors clarify whether the loss is calculated in euclidean space or in hyperbolic space?
- Why does nnUnet fail with low contrast augmentation strength such that it performs random (0 dice score)? It looks like an implementation issue. This seems consistent across datasets (Figure A1-A2)

References:

[1] Nickel, Maximillian, and Douwe Kiela. "Learning continuous hierarchies in the lorentz model of hyperbolic geometry." International conference on machine learning. PMLR, 2018.

---

> ### Author Response · Authors · 2026-01-25
>
> We thank the reviewer for their constructive feedback. We believe we have addressed the questions and suggestions below:
>
> **Why is hyperbolic geometry so effective for noisy robustness?**
> We thank the reviewer for the suggestion to better motivate the role of geometry in noise robustness and to form an explicit hypothesis. We have updated our manuscript with a more explicit hypothesis and discussion on the geometric properties of hyperbolic spaces that induce robustness to noise, along with an additional experiment (Appendix J) to support this analysis.
>
> A key property of hyperbolic space is that volumes grow exponentially with norm, instead of polynomially. As a result, classes are more strongly separated and noise perturbations are less likely to cause pixel representations to cross class boundaries. This makes hyperbolic geometry highly suited for dealing with noise. We have validated this hypothesis by comparing Hyperbolic and Euclidean U-Net across datasets and noise variants. In Appendix J, we shows that Hyperbolic U-Net consistently achieves higher class separation ratios.
>
> **Novelty and utility of hyperbolic bilinear upsampling**
> We apologize for any confusion regarding our hyperbolic bilinear upsampling. Indeed, for the experiments in Table 1 we use our hyperbolic transpose convolutions. However, transpose convolutions are inherently costly and even more so in hyperbolic space. Hyperbolic transpose convolutions lead to GPU memory exhaustion for Hyperbolic U-Net++, whereas hyperbolic bilinear upsampling remains viable. This upsampling was therefore used in Table D1. We have better clarified in the manuscript what the pros and cons of our two approaches are and when which approach is used in our experiments.
>
> Regarding the **geodesic interpolation** of the hyperbolic bilinear upsampling: this operation is fully on the manifold since both mappings have the same origin $\boldsymbol{a}$. Parallel transport is only required when tangent vectors need to be moved between different base points on the manifold, which is not the case in our approach.
>
> **Curvature of the hyperbolic UNet**
> We have experimented with both fixed and trainable curvature settings. We did not observe a strong difference in training stability or segmentation performance, with a slightly better performance for a learnable curvature initialized with $c = 0.1$. In our experiments, the learned curvature values typically converges to the range [0.1, 1.5]. We have included the ablation study in Appendix K.
>
> **Hyperbolic UNet architecture**
> Skip connections in the Hyperbolic U-Net are implemented using the Poincaré concatenation operation. The overall architecture replicates the standard Euclidean U-Net, with all layers replaced by their hyperbolic counterparts. We do not apply an explicit clamping or projection step between layers. In practice, we did not observe embeddings approaching the boundary of the Poincaré ball or training instabilities related to norm growth. We have added a new Appendix L detailing the full Hyperbolic U-Net architecture and skip-connection design. We use the Poincaré model due to its widespread use and available implementation in the hypll library.
>
> **Optimizer, loss and init**
> We use Riemannian Adam for all our experiments and for the Euclidean baselines, Adam. We have updated the implementation details in Section 4 accordingly. In all experiments, the same loss (Dice-Focal loss) is used for both Euclidean and Hyperbolic U-Net, as the loss is applied on softmax probability outputs, which is independent of the underlying geometry of the feature representations. In Table P1 (Appendix P), we also compare our initialization to the baselines, demonstrating better performance.
>
> **nnU-Netv2 framework**
> We clarify that all baseline comparisons in our experiments are performed using the latest nnU-Net implementation, corresponding to nnU-Netv2. Our experiments are based on the official nnU-Net repository, which implements the v2 framework. The repository recommends citing the original nnU-Net paper. For clarity, we mention nnU-Netv2 explicitly now in the setup and added the reference mentioned by the reviewer.
>
> **Noise augmentation**
> Hyperbolic U-Net is trained without artificial noise augmentation, whereas the nnU-Netv2 baseline includes standard noise-based data augmentation. Despite this, Hyperbolic U-Net outperforms nnU-Netv2 with augmentation, highlighting the potential of hyperbolic geometry.
>
> **nnU-Netv2 performance in low contrast setting**
> We agree that this behavior warrants clarification. Under the low contrast augmentation setting, nnU-Netv2 consistently predicts the background class for all pixels, resulting in empty foreground predictions and therefore a Dice score of 0. This is reproducible across datasets (Figures A1–A2) and does not arise from differences in inference code or evaluation. We believe it is related to nnU-Net's sensitivity to contrast degradation under this specific augmentation setting.

---

> > ### Comment · Reviewer_9Q83 · 2026-01-26
> >
> > Thanks for the detailed answer. I have 2 following questions. First of all, my questions related to implementation (i.e. concatenation and optimizer) are clarified, but maybe the authors realize as well, the code is very important for hyperbolic papers, currently provided github repo is empty, I suggest authors to realize the code before the final decision. Secondly, the larger representation capacity of the hyperbolic spaces does not point a well grounded reason for robustness against noise, especially given that the norms are constrained to avoid numerical instabilities and proposed norm preserving initialization (expmap uses the input norm). In my opinion, current reasoning can be solidified by rephrasing it as adversarial attack or OOD, since there is no noise augmentation during training.
> >
> > Additionally, authors replied that the softmax is based on probability there is no underlying geometry. But still the final logits are in hyperbolic space, how are they transformed into probabilities?

---

> > > ### Author Response · Authors · 2026-01-27
> > >
> > > We thank the reviewer for their guidance and constructive suggestions, please see below for our answer to your questions:
> > >
> > > **Code availability.** We fully agree with the reviewer. We have updated the GitHub repository with a fully functional implementation, including training and evaluation scripts, along with a detailed README to help reproducibility. We also plan to release the pre-trained models.
> > >
> > > **Robustness interpretation.** While there are indeed regularizations in place, we note that the observed increased capacity and larger distances to hyperplanes occurs in our actual implementations, hence it is not just a theoretical benefit. We thank the reviewer for the suggestion to solidify the reasoning by rephrasing noise robustness as adversarial attack or OOD. We agree that these topics are linked, as all focus on dealing with shifts in samples and/or distributions. Several works have already shown that hyperbolic learning positively impacts adversarial robustness (van Spengler et al., 2024) and out-of-distribution detections (Gonzalez-Jimenez et al., 2024). We will update the paper with this link and interpretation.
> > >
> > > **From hyperbolic features to probabilities.** In all experiments, the loss is computed entirely in the Euclidean output space. While both the encoder and decoder operate in hyperbolic space, the final prediction layer explicitly maps the hyperbolic features to Euclidean logits (as described in Section 3.1). These Euclidean logits are then transformed into probabilities via a standard softmax operation and optimized using a Dice–Focal loss against the ground-truth segmentation masks. As a result, the loss function and probability computation are independent of the underlying geometry of the intermediate feature representations and are identical for both Euclidean and hyperbolic variants of the model. This design follows the formal definition of hyperbolic neural networks introduced by Shimizu et al. (2021), in which all learnable parameters of the representation learning components reside in hyperbolic space, while predictions are obtained via an explicit mapping to a Euclidean output space.

---

> > > > ### Comment · Reviewer_9Q83 · 2026-01-28
> > > >
> > > > I still cant see how the exponential volume increase of hyperbolic spaces connects to robustness to test time augmentation, but given the responses to the other issues, I will increase my score

---

### Official Review · Reviewer_YcMC · 2026-01-06

**Confidence:** 4
**Preliminary Rating:** 4
**Final Rating:** 5

**Summary:**

This work addresses improving noise robustness in medical image segmentation by introducing a hyperbolic U-Net whose operations are performed in a hyperbolic space. Building on previous work in hyperbolic geometry for neural networks (hyperbolic convolutional neural networks), especially the Poincaré ball model and Poincaré hyperplane, the authors proposed hyperbolic bilinear upsampling, an essential technique combined with Poincaré transposed convolution to realise decoder processing in a hyperbolic U-Net. They also proposed a Newton-scaled weight initialisation method that maintains the input norm in the hyperbolic U-Net architecture's output. The authors demonstrated the validity of the proposed method using 7 publicly available datasets.

**Strengths:**

- Noise robustness in segmentation is an essential factor for medical image processing.
- The submitted manuscript has a high readability. The structure and presentation of the manuscript are solid.
- The authors introduced a new operation (hyperbolic bilinear upsampling) and integrated it into Poincaré convolution and Poincaré transposed convolution for presenting the hyperbolic U-net.
- The authors also proposed a new initialisation method for the hyperbolic U-net.
- The proposed methods are based on a kind of Riemann geometry and sound mathematically solid if their derivations are correct.
- The proposed method achieved noise robustness against Gaussian, Poisson, and Rician noises and brightness and contrast changes.

**Weaknesses:**

- The mathematical derivation process of the proposed techniques is missing. The current descriptions of hyperbolic geometry, especially transposed convolution and bilinear upsampling, have several gaps among definitions.
- Introductory explanations of basic Riemann geometry are missing. Without basic knowledge about Riemann geometry, it is a little bit hard to understand Section 3 for a reader in medical image processing.

**Detailed Comments:**

The authors thought that deep learning in hyperbolic space has strong potential for robustness against noise and appearance changes, such as brightness and contrast changes, and proposed the hyperbolic U-Net. To present a hyperbolic U-Net, the authors introduced a new operation (hyperbolic bilinear upsampling) and integrated it into Poincaré convolution and Poincaré transposed convolution. For training the hyperbolic U-Net, they also  proposed a new initialisation method. The presented experimental evaluations, using 7 publicly available datasets and 5 noise types, convincingly demonstrate the validity of the proposed hyperbolic U-Net and its high robustness against noise. Totally, this submission looks good work.

**Justification Of Final Rating:**

Based on the review comments, the authors responded to all my comments, and their responses are convincing. The authors’ new techniques: hyperbolic bilinear upsampling and Newton-scaled weight initialisation, achieved the purpose of this work, i.e. noise robustness while keeping competitive performance for noise-absent data against the Euclidean U-net. Since the noise problem in the input is an essential issue, this work has the potential to contribute to medical image processing. Furthermore, this work is based on mathematical derivations and builds on solid previous works. Moreover, in the revised version, the authors added a geometric explanation of nose robustness. Experimental evaluations are also convincing. As a result, I conclude that this work should be accepted for the MIDL presentation this year.

**Justification Of The Preliminary Rating:**

As I commented in "Strengths" and "Detailed Comments", this work has several outcomes for medical image processing. Even though they demonstrate the experimental validity of the proposed approach, the derivations/proofs of the proposed new definitions should be presented in Appendices to clarify their validity, since this work is based on the mathematical definitions of Riemann geometry and extends them.

**Questions To Address In The Rebuttal:**

How did the authors derive the hyperbolic bilinear upsampling and Newton-scaled weight initialisation? Please add derivation processes/proofs to present their validity as Appendices. At the definition of \gamma, what does the operator \cdot mean? If it represents just a scalar multiplication,  \cdot should be removed. Furthermore, here, exp and log should be Roman style.

---

> ### Author Response · Authors · 2026-01-25
>
> We thank the reviewer for their feedback and for the positive comments on the paper. We have addressed the questions below:
>
> **Extending mathematical derivations.**
> We thank the reviewer for pointing out the need for additional introductory explanations of basic Riemannian geometry. We have extended our manuscript with a paragraph in the beginning of Section 3 (Methods), to include more background on hyperbolic geometry. The new additions are highlighted in red in the revised version of the manuscript.
>
> **Extending mathematical derivations.**
> We have expanded Section 3.2 (Poincaré Transposed Convolutions) to provide more details regarding the proposed method. We have also added the derivations for the hyperbolic bilinear upsampling and Newton-scaled weight initialisation in Appendices H and I, respectively.
>
> **Textual revisions.**
> Regarding the definition of $\gamma$, the operator $\cdot$ indeed means scalar multiplication. We have removed the $\cdot$ for clarity. We have made the corrections for scalar multiplication throughout the revised manuscript. The exp and log in the function $\gamma$ have also been updated in Roman style. Thank you.

---

> > ### Comment · Reviewer_YcMC · 2026-01-29
> >
> > Thank you for the feedback. The authors fully replied to my comments. Many thanks.

---

### Author Rebuttal · Authors · 2026-01-25

**Rebuttal:**

We attach the revised version of our manuscript, which has been thoroughly updated in response to the reviewers’ constructive comments. We sincerely thank all the reviewers for their feedback and suggestions, which have helped us improve both the clarity and rigor of our work. In this revision, all new or substantially revised text is highlighted in red so that readers can easily identify the changes and follow our updates. We believe that these enhancements address the key concerns raised during the review process and further strengthen our paper.

**Supporting Material:**

/attachment/22f2f466681bb4600aab144907683c2e599d024b.pdf

---

### Meta-Review · Area_Chair_TejZ · 2026-02-06

**Recommendation:** Accept (Oral)
**Confidence:** 4

**Metareview:**

Two reviewers recommend strong acceptance, one reviewer rates this paper borderline but leans towards acceptance.
All reviewers agree the rebuttal addresses most of their concerns and clearly improved the paper.

---

### Decision · Program_Chairs · 2026-02-13

Accept (Poster)